# Scalable high performance radio frequency electronics based on large domain bilayer MoS$_2$

Qingguo Gao[1], Zhenfeng Zhang[1], Xiaole Xu[1], Jian Song[1], Xuefei Li[1] & Yanqing Wu [1]

Atomically-thin layered molybdenum disulfide (MoS$_2$) has attracted tremendous research attention for their potential applications in high performance DC and radio frequency electronics, especially for flexible electronics. Bilayer MoS$_2$ is expected to have higher electron mobility and higher density of states with higher performance compared with single layer MoS$_2$. Here, we systematically investigate the synthesis of high quality bilayer MoS$_2$ by chemical vapor deposition on molten glass with increasing domain sizes up to 200 μm. High performance transistors with optimized high-$\kappa$ dielectrics deliver ON-current of 427 μA μm$^{-1}$ at 300 K and a record high ON-current of 1.52 mA μm$^{-1}$ at 4.3 K. Moreover, radio frequency transistors are demonstrated with an extrinsic high cut-off frequency of 7.2 GHz and record high extrinsic maximum frequency of oscillation of 23 GHz, together with gigahertz MoS$_2$ mixers on flexible polyimide substrate, showing the great potential for future high performance DC and high-frequency electronics.

[1] Wuhan National High Magnetic Field Center and School of Optical and Electronic Information, Huazhong University of Science and Technology, Wuhan 430074, China. These authors contributed equally: Qingguo Gao, Zhenfeng Zhang. Correspondence and requests for materials should be addressed to Y.W. (email: yqwu@mail.hust.edu.cn)

Two-dimensional (2D) semiconductors have received great research attention for applications in the emerging field of ubiquitous electronics, such as sensors, memory, and logic applications owing to their atomically thin body and excellent carrier transport properties[1–9]. Flexible electronics in wireless communication is one of the most promising field which has witnessed rapid development of flexible passive components and active components[10]. However, despite tremendous interest in graphene transistors for active radio frequency (RF) components[11,12], it still remains a challenging issue that the gapless nature of graphene gives rise to poor current saturation and large output conductance in these transistors, which are detrimental for amplifying and mixing high frequency signals. Recently, great progress has been made on high frequency transistors and circuits based on 2D transition metal dichalcogenides, such as molybdenum disulfide ($MoS_2$), where the key disadvantage of graphene can be overcome[13–17]. Mechanically exfoliated $MoS_2$ on quartz substrates has shown high extrinsic radio frequency performances[13]. In order to provide a low-cost scalable solution, large-area synthesis of $MoS_2$ atomic films by chemical vapor deposition (CVD) was developed with progressive improvement by many research groups[18–25]. Recently, RF transistors on flexible polyimide substrates based on monolayer $MoS_2$ grown by CVD exhibited an extrinsic cut-off frequency $f_T$ of 2.7 GHz and maximum oscillation frequency $f_{max}$ of 2.1 GHz[16] and, furthermore, an extrinsic $f_T$ of 3.3 GHz and $f_{max}$ of 9.8 GHz were demonstrated using an embedded gate structure on $SiO_2$/Si substrates[17]. However, these parameters are still well below the devices based on exfoliated $MoS_2$, severely limiting their high frequency applications. It is well known that the carrier mobility of bilayer $MoS_2$ is higher than that of monolayer and, as a result, better performance can be obtained owing to the higher density of states and smaller bandgap which is more suitable for high frequency electronics[26,27]. However, bilayer $MoS_2$ growth by CVD suffers from small domain sizes and poor mobility, restricting its device performance[27–30].

Here, high mobility large domain bilayer $MoS_2$ growth by CVD on molten glass is realized by adjusting the weight of $MoO_3$ precursor during growth. The largest domain size of 200 μm can be obtained and the resulting single-crystal triangular bilayer $MoS_2$ demonstrates a room temperature electron mobility of 36 $cm^2 V^{-1} s^{-1}$. A back-gated $MoS_2$ transistor with 40 nm channel length exhibits a record high ON-current ($I_{on}$) of 1.52 mA μm$^{-1}$ at 4.3 K with optimized high-$\kappa$ dielectrics. State-of-the-art RF transistors based on bilayer $MoS_2$ are demonstrated with a record high extrinsic cut-off frequency $f_T$ of 7.2 GHz and maximum oscillation frequency $f_{max}$ of 23 GHz[15–17]. Moreover, $MoS_2$ RF transistors and frequency mixers on flexible substrates are demonstrated with a $f_T$ of 4 GHz and $f_{max}$ of 9 GHz where the mixer remains functional in gigahertz regime.

## Results

**Material synthesis and characterization.** Bilayer $MoS_2$ was grown on molten glass by the vapor-phase reaction of sulfur and $MoO_3$ in a thermal CVD system. Schematic view of the CVD setup is shown in Fig. 1a. The CVD growth process was carried out at ambient pressure where the temperatures of sulfur powders and $MoO_3$ during growth were kept at 230 and 830 °C, respectively. Optical microscopy images of the resulting $MoS_2$ domains on the molten glass with increasing weight of $MoO_3$ are shown in Fig. 1b–e, where well-defined triangular shapes and clear uniform color contrast of CVD bilayer $MoS_2$ are grown (for more details see Methods). As shown in Fig. 1b, monolayer triangular $MoS_2$ domains tend to grow when the weight of $MoO_3$ is less than 1 mg. As the weight of $MoO_3$ slowly increases from 1.5 to 6 mg, bilayer

$MoS_2$ starts to grow with an increasing domain size for the same growth duration as shown in Fig. 1c–e. And, with the $MoO_3$ weight of 6 mg, the largest domain size up to 200 μm is obtained. The shrinking size of bilayer $MoS_2$ compared with the monolayer underneath may be due to the first layer has a faster growth rate than the second layer[31], and the growth time decreases from the first to second layer[27]. This is the largest bilayer $MoS_2$ domain among reported results to the best of our knowledge (Supplementary Table 1)[27,29,30]. This result shows that the mass-transport process controls the growth of bilayer $MoS_2$ and there is a positive linear relationship between the growth rate and the weight of the $MoO_3$ (detailed discussions in Supplementary Note 1) and the advantage of using molten glass as growth substrate (Supplementary Note 2). More optical images and scanning electron microscope (SEM) images are shown in Supplementary Note 3. As shown in Fig. 1f–h, the film thickness measured by atomic force microscope (AFM) of the monolayer and bilayer CVD $MoS_2$ is around 0.72 nm and 1.34 nm, respectively.

Raman spectroscopy is widely used to distinguish between monolayer and bilayer $MoS_2$ based on the spectral position of the characteristic $E_{2g}^1$ and $A_{1g}$ peaks[32]. Figure 2a shows the typical Raman spectra of monolayer and bilayer $MoS_2$ after being transferred onto $SiO_2$/Si substrates. The delta values between the $E_{2g}^1$ and $A_{1g}$ peaks of monolayer and bilayer $MoS_2$ are 18.9 and 22.4 cm$^{-1}$, respectively, consistent with previous reports[30,32]. Figure 2b, c show Raman intensity mappings recorded at 385 cm$^{-1}$ and 405 cm$^{-1}$, respectively. Bilayer $MoS_2$ region has a higher intensity of Raman signal than that of monolayer, and the uniform contrast indicates a good uniformity of the bilayer film. Figure 2d compares the typical photoluminescence (PL) spectra of the monolayer and bilayer $MoS_2$, where the peaks corresponding to the $A1$ and $B1$ direct exciton transitions with the energy split from valence band spin–orbital coupling[33]. The PL intensity of the bilayer $MoS_2$ is about 60% lower than monolayer because of the transition from direct bandgap in monolayer to the indirect bandgap in bilayer. Figure 2e shows the PL intensity mapping of the bilayer domain recorded at 1.85 eV, further confirming the good uniformity of the bilayer $MoS_2$. Transmission electron microscopy (TEM) and electron diffraction studies were performed to confirm the single crystalline nature and to determine the lattice structures of the bilayer $MoS_2$ domains. Figure 2f shows the low-resolution TEM image of a CVD $MoS_2$ domain transferred on copper grids. A magenta dotted line is used to indicate the boundary between monolayer and bilayer $MoS_2$ where the left side of the line is monolayer and the right side is bilayer. Selected area electron diffraction (SAED) was conducted at location 1 in Fig. 2f and the diffraction pattern is shown in Fig. 2g. These diffraction peaks yield (100) direction with a lattice plane spacing of 2.83 Å. SAED images on another four selected openings with same electron diffraction patterns are shown in Supplementary Note 4, confirming the uniform crystallinity of the bilayer $MoS_2$ domain. High-resolution TEM (HRTEM) analysis was also performed to evaluate the quality and crystallinity of the $MoS_2$ films on the atomic scale, as shown in Fig. 2h, revealing the AA stacking order of the bilayer $MoS_2$ domain. As depicted in Fig. 2i, a thickness of 1.26 nm can be determined from an HRTEM image recorded from a folded edge of the bilayer $MoS_2$ domain and consistent with the thickness of bilayer $MoS_2$[28].

**DC characterizations on back-gated devices.** To characterize the electronic properties of bilayer $MoS_2$, back-gated field-effect transistors (FETs) with channel lengths from 3 μm down to 40 nm were fabricated on HfLaO substrates with a Si back gate (details of $MoS_2$ transfer onto silicon substrate are discussed in Supplementary Note 5). High-$\kappa$ dielectrics provide a better interface with the 2D semiconductors channel with smaller effective oxide thickness,

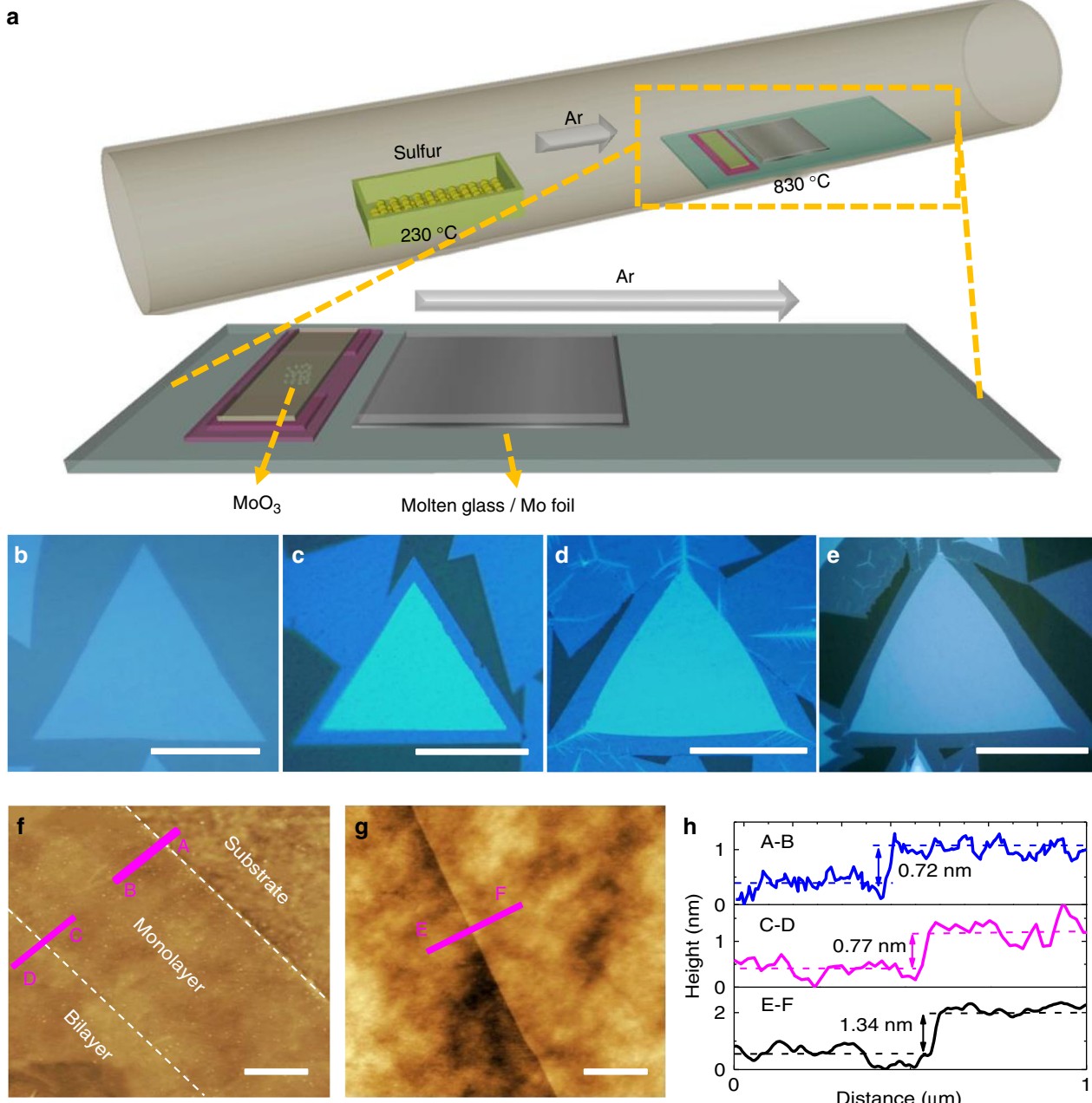

**Fig. 1** Bilayer MoS$_2$ synthesis on molten glass and morphology characterization. **a** Schematic of the CVD setup for the synthesis of bilayer MoS$_2$ on molten glass. **b–e** Optical micrographs of CVD grown MoS$_2$ on molten glass; the corresponding weight of MoO$_3$ are 1, 1.5, 3, and 6 mg, respectively. Scale bars are 30, 40, 50, and 100 μm, respectively. **f, g** AFM images of bilayer MoS$_2$ on SiO$_2$/Si substrates after transfer. Scale bars are 1 μm. **h** AFM data of line cut A–B, C–D, and E–F corresponding to the thickness of monolayer MoS$_2$, the height difference of bilayer and monolayer MoS$_2$, the thickness of bilayer MoS$_2$, respectively. Panels **f, g** located at the edge and crack in the middle of bilayer MoS$_2$ flake, respectively

which help to improve the output performance of the transistors[34,35] (details of the high-$\kappa$ dielectrics are discussed in Supplementary Notes 6, 7). Optical microscope images and the corresponding SEM images of the back-gated MoS$_2$ transistors are shown in Fig. 3a, b. Transfer characteristics of the transistors in the linear region based on monolayer and bilayer MoS$_2$ with the same channel length of 3 μm are plotted in Fig. 3c. The current and transconductance are more than 50% higher in the bilayer MoS$_2$ devices compared with the monolayer devices. Note here that the two devices were made on the same substrate with the same oxide thickness and fabrication process to remove processing-induced differences between the two cases. Output characteristics of the

same 3 μm channel bilayer MoS$_2$ device at 300 K and 4.3 K are shown in Fig. 3d, e, respectively. The output drain current increases from 35 μA μm$^{-1}$ at 300 K to 65 μA μm$^{-1}$ at 4.3 K, an improvement of over 80% compared to room temperature. Intrinsic field-effect mobility of bilayer FETs is calculated to be 36 and 127 cm$^2$ V$^{-1}$ s$^{-1}$ at 300 K and 4.3 K, respectively (details of the mobility calculations are discussed in Supplementary Note 8). The detailed temperature dependence of mobility is plotted in Fig. 3f, showing a steady increase of mobility with decreasing temperature, which can be mainly attributed to the reduced phonon scattering and the temperature dependence coefficient is consistent with previous work shown in Supplementary Table 2[6] (details of the

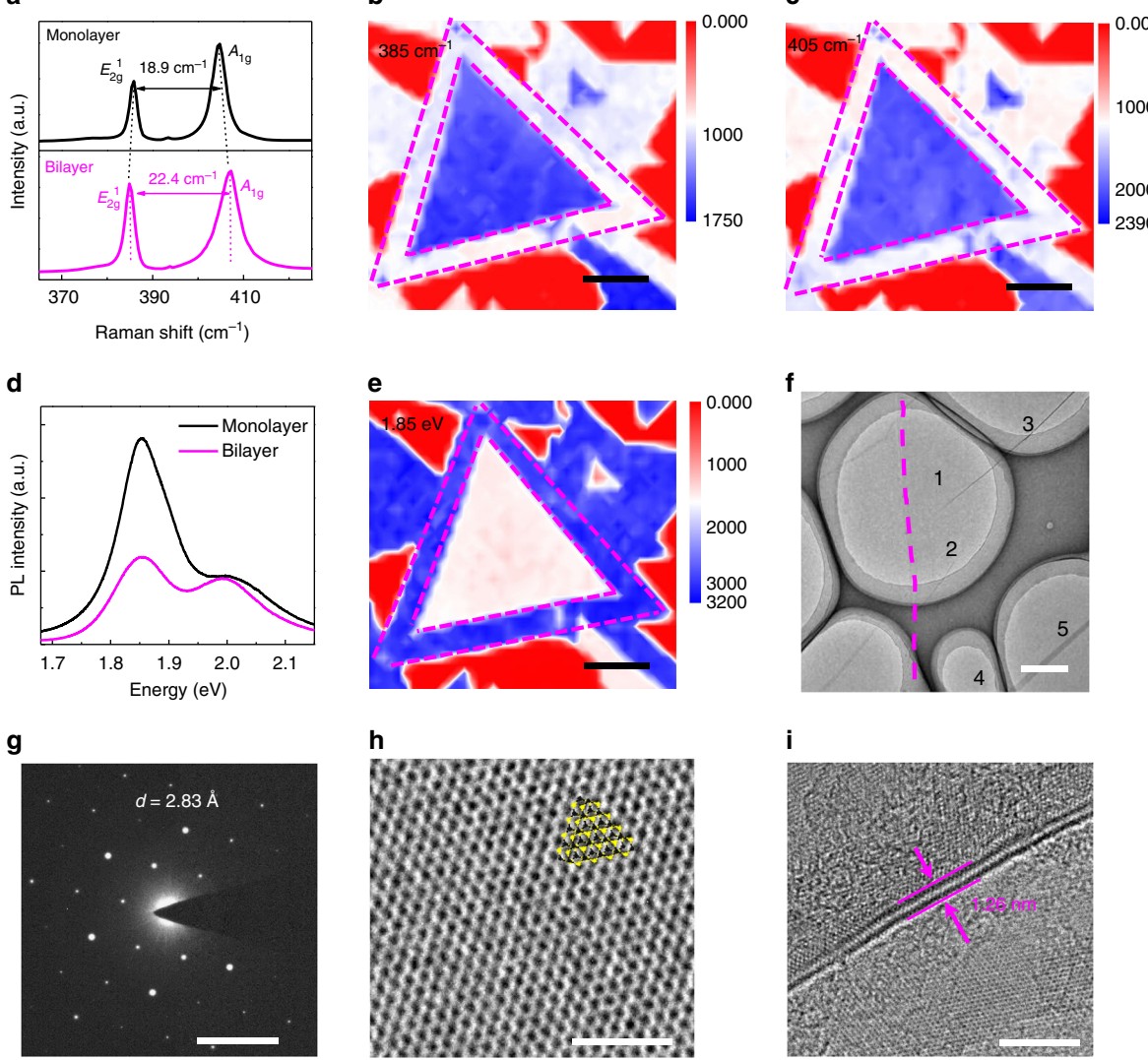

**Fig. 2** Material imaging and crystal structure characterization. **a** Raman spectra of CVD monolayer (black line) and bilayer $MoS_2$ (magenta line). For monolayer one, the $E_{2g}^1$ peak is at 385.8 cm$^{-1}$ and the $A_{1g}$ peak is at 404.7 cm$^{-1}$, corresponding to a delta of 18.9 cm$^{-1}$. For bilayer $MoS_2$, the $E_{2g}^1$ peak is at 384.8 cm$^{-1}$ and the $A_{1g}$ peak is at 407.2 cm$^{-1}$, corresponding to a delta of 22.4 cm$^{-1}$. **b**, **c** Raman spectroscopy maps of the peaks at 385 cm$^{-1}$ and 405 cm$^{-1}$. Scale bars are both 20 μm. **d** Photoluminescence spectra of CVD monolayer (black line) and bilayer (magenta line) $MoS_2$. The peaks corresponding to the $A1$ and $B1$ direct excitonic transitions with the energy split from valence band spin–orbital coupling. **e** PL spectroscopy map of the peak at 1.85 eV. Scale bar is 20 μm. **f** Low-resolution TEM image of the boundary between bilayer (right) and monolayer (left). Scale bar is 1 μm. **g** Typical SAED pattern image in the bilayer region in **f**. Scale bar is 5 nm$^{-1}$. **h** HRTEM image and atomic configuration of AA stacked bilayer $MoS_2$. Scale bar is 2 nm. **i** HRTEM image recorded from the folded edge of bilayer $MoS_2$. Scale bar is 5 nm

mobility analysis are discussed in Supplementary Note 9). As can be seen in Supplementary Table 1, the mobility of bilayer $MoS_2$ growth on the molten glass in this work shows a clear improvement over the previous results[9,27,28,30]. Transfer characteristics at the linear region from transistors with reducing channel lengths down to 40 nm are shown in Fig. 3g, where the drain current increases as the channel length decreases (channel length-dependent output characteristics are discussed in Supplementary Note 10). Output characteristics of the 40 nm devices are measured at 300 K with 427 μA μm$^{-1}$ ON-current and at 4.3 K with maximum $V_{gs}$ up to 6 V where an ON-current of 1.52 mA μm$^{-1}$ can be achieved as shown in Fig. 3h. This is the largest drive current of $MoS_2$ transistors reported thus far[36–38]. We attribute this high ON-current at 300 K and 4.3 K to the optimized interface quality, better electrostatic control, high-$\kappa$ doping effect by HfLaO[34,35,39–43], and mobility boost at low temperatures[6].

**Top-gate high frequency transistors.** Top-gated two-finger RF transistors based on bilayer $MoS_2$ have been fabricated and shown in the schematic view in Fig. 4a, with different gate lengths of 90, 190, and 300 nm and the same gate width of 30 μm. Figure 4b shows SEM images of a device with a gate length of 90 nm, exhibiting the precise alignment of gate structure to the source/drain area. There is no overlap in our device design to avoid excess gate to source capacitance $C_{gs}$ and gate to drain capacitance $C_{gd}$. At the same time, the gate to drain access length $L_{gd}$ and gate to source access length $L_{gs}$ is minimized to decrease the series resistance. DC characterizations of CVD bilayer $MoS_2$ RF transistor is shown in Supplementary Note 11. Standard on-chip S-parameter measurements up to 30 GHz are used for RF measurement with Lakeshore probe station (for more details see Methods). Figure 4c–e show the as-measured extrinsic short-circuit current gain ($|h_{21}|$), Mason's unilateral power gain ($U$),

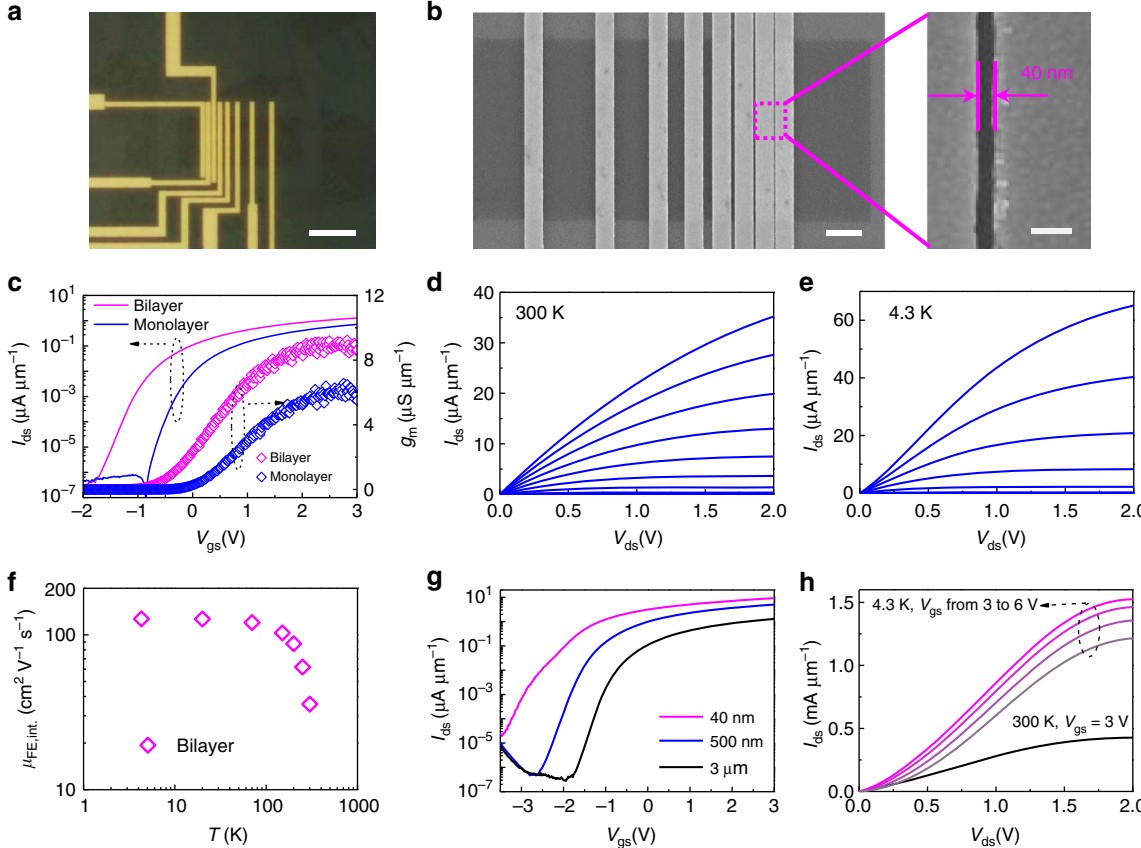

**Fig. 3** DC electrical characterization of back-gated bilayer $MoS_2$ transistors at room temperature and low temperatures. **a** Optical micrograph of back-gated $MoS_2$ transistors with different channel lengths. Scale bar is 10 μm. **b** The corresponding SEM images of the active device region and the zoom-in picture. Scale bar are 2 μm and 100 nm, respectively. **c** The $I_{ds} - V_{gs}$ transfer characteristics at 50 mV bias voltage and $g_m - V_{gs}$ curves at 1 V bias voltage for the 3 μm channel length back-gated monolayer (blue line and open diamonds, respectively) and bilayer (magenta line and open diamonds, respectively) $MoS_2$ transistors. **d, e** The $I_{ds} - V_{ds}$ output characteristics of the bilayer $MoS_2$ transistor at 300 K and 4.3 K. The back-gate voltages vary from $-1$ to 3 V with a step of 0.5 V. **f** The extracted intrinsic field-effect mobility versus temperature of bilayer $MoS_2$ FETs. **g** The $I_{ds} - V_{gs}$ transfer characteristics at 50 mV bias voltage at room temperatures for different channel lengths of 40 nm (magenta line), 500 nm (blue line), and 3 μm (black line). **h** The $I_{ds} - V_{ds}$ output curves at 300 K and 4.3 K for bilayer $MoS_2$ transistors with a channel length of 40 nm. A record $I_{on}$ of 1.52 mA μm$^{-1}$ was achieved

and voltage gain as a function of frequency for the 90 nm device (details on $f_T$ and $f_{max}$ are discussed in Supplementary Note 12). As shown in Fig. 4c, the extrinsic cut-off frequency $f_T$ derived from the short-circuit current gain is 7.2 GHz, the highest extrinsic $f_T$ achieved for CVD $MoS_2$[17] and is consistent with the values extracted from Gummel's method (Supplementary Note 13). While the cut-off frequency $f_T$ defines the frequency at which short-circuit current gain becomes unity, the maximum oscillation frequency $f_{max}$ is defined as the frequency at which Mason's unilateral power gain equals unity. This figure of merit is more relevant in terms of power amplifying[44]. Figure 4d shows the unilateral power gain versus frequency with an extrinsic $f_{max}$ of 23 GHz, which is 2.3 times greater than the previously reported extrinsic $f_{max}$ for the CVD $MoS_2$, and is also the highest value in all reported 2D semiconductors as shown in Supplementary Table 3[17,45]. In this work, the improvement of $f_T$ and $f_{max}$ are attributed to the CVD bilayer $MoS_2$ with high carrier mobility and low contact resistance, short gate lengths designed, and the high output resistance. As shown in Supplementary Note 14, intrinsic de-embedded $f_{T,int}$ of 78 GHz and $f_{max,int}$ of 34 GHz are obtained, and saturation velocity of $4.4 \times 10^6$ cm s$^{-1}$ can be obtained from the intrinsic $f_T$. Voltage gain Av extracted as $Z_{21}/Z_{11}$ is also an important parameter for $MoS_2$ RF transistors[15,17] and, as shown in Fig. 4e, the extrinsic voltage gain is positive up to 4.2 GHz. To demonstrate the overall performance of CVD

bilayer $MoS_2$ RF transistors, cut-off frequencies and maximum oscillation frequencies with different gate lengths are plotted in Fig. 4f, g, respectively. Both $f_T$ and $f_{max}$ increase as the gate lengths decrease, and this positive scaling is benefited from the low contact resistance of bilayer $MoS_2$ devices especially in short channel devices (detailed RF characteristics of 190 nm and 300 nm devices are shown in Supplementary Note 15). It is well known that the high output conductance of graphene RF transistors due to the lack of bandgap typically results in an unsatisfactory $f_{max}/f_T$ ratio. Figure 4h shows the calculated $f_{max}/f_T$ ratio from the bilayer $MoS_2$ RF transistors where high $f_{max}/f_T$ ratios up to three are obtained. This can be attributed to the improved output conductance of bilayer $MoS_2$ transistors[46].

**Gigahertz frequency mixers on flexible substrates**. A frequency mixer is a key component of RF systems and is widely used in wireless communications. Currently, the demonstrated frequency mixers based on $MoS_2$ mainly work in MHz regime, mainly due to the relatively low extrinsic high frequency performance of reported $MoS_2$ RF transistors[15,16]. Based on the RF transistors shown above, frequency mixers were measured where RF and local oscillator (LO) signals were combined, biased via a bias-Tee, and applied to the gate of the device. The intermediate frequency (IF = RF − LO) signal was measured with a signal analyzer as shown in Fig. 5a. Figure 5b shows the output spectrum of the

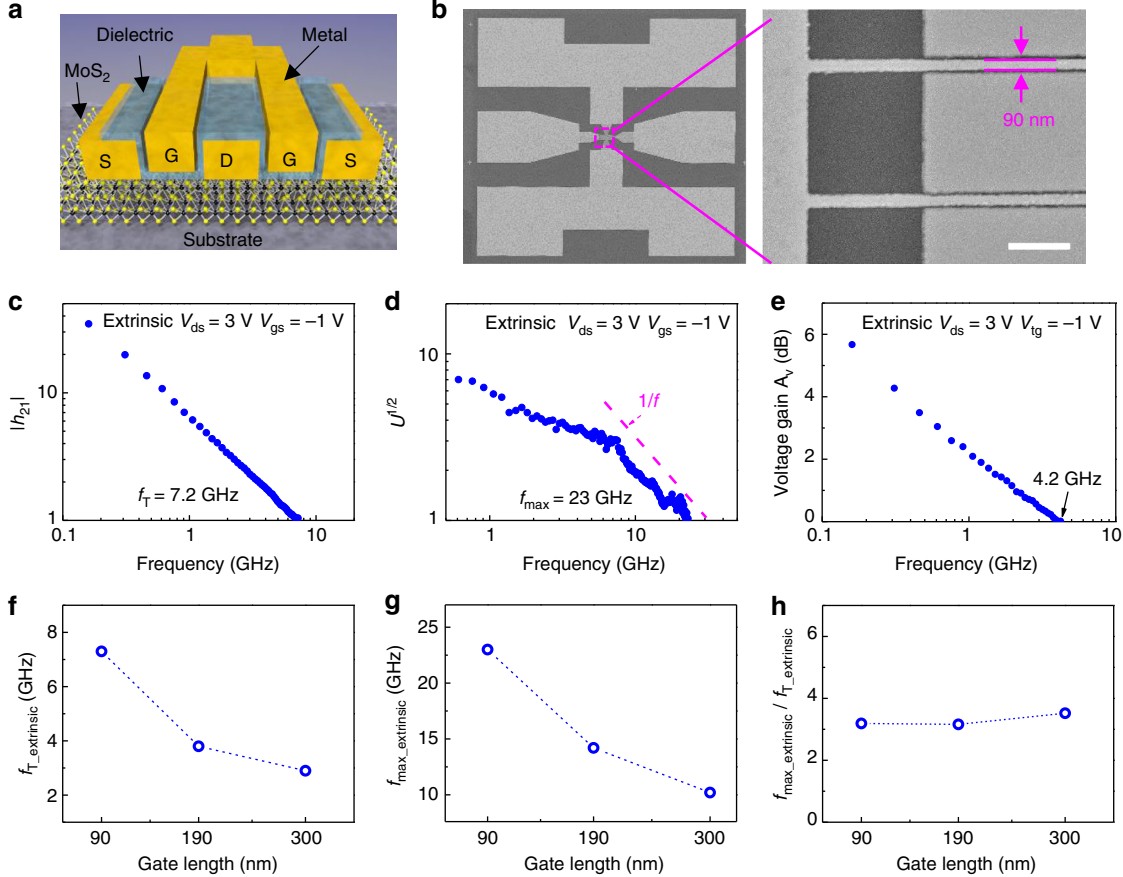

**Fig. 4** High frequency measurement of short channel RF transistors. **a** Schematic illustration of bilayer MoS$_2$ RF transistor. S, source, D drain, G gate. **b** The SEM images of MoS$_2$ RF transistor with dual-channel structure show excellent alignment of gate to source and drain. Scale bar is 500 nm. **c** Small-signal current gain $|h_{21}|$ versus frequency for device with gate length of 90 nm. The extrinsic cut-off frequencies $f_T$ is 7.2 GHz. Where $V_{ds} = 3$ V and $V_{gs} = -1$ V. **d** Unilateral power gain $U$ versus frequency for device with gate length of 90 nm. The extrinsic maximum oscillation frequency $f_{max}$ is 23 GHz. Where $V_{ds} = 3$ V and $V_{gs} = -1$ V. **e** Voltage gain versus frequency for device with gate length of 90 nm. Where $V_{ds} = 3$ V and $V_{gs} = -1$ V. **f** Extrinsic $f_T$ as a function of gate length. **g** Extrinsic $f_{max}$ as a function of gate length. **h** $f_{max}/f_T$ as a function of gate length

configured MoS$_2$ mixer in the gigahertz range, where an IF signal of $f_{IF} = 100$ MHz is clearly seen with $f_{RF} = 1.5$ GHz and $f_{LO} = 1.4$ GHz. The conversion gain versus the applied LO power is plotted in Fig. 5c, showing higher conversion gain can be achieved with increasing LO power from 3 to 9 dBm, consistent with previous work[47]. A conversion gain of $-30.7$ dB was obtained at the LO power of 9 dBm. It should be noted that conversion gain here is defined as the ratio between the IF output signal power and the RF input signal power[48].

The 2D semiconductors have received high expectations in flexible electronics due to their ultrathin body nature and RF devices are essential for analog signal transmitting, amplifying and processing in those applications. As a result, we fabricated high frequency bilayer MoS$_2$ transistors on flexible polyimide films from Dupont using the same fabrication and measurement techniques, where the DC characteristics of a representative flexible transistor can be found in Supplementary Note 16. An extrinsic cut-off frequency $f_T$ of 4 GHz and maximum oscillation frequency $f_{max}$ of 9 GHz are achieved in a 300 nm gate length device as shown in Fig. 5d, showing significant improvement over previous results based on monolayer CVD MoS$_2$ on flexible polyimide substrates[16]. The RF characteristics of the transistors after various bending conditions can be found in Supplementary Note 17. Moreover, we also constructed a gigahertz MoS$_2$ RF mixer on flexible substrates with the same test setup as on rigid substrates. The RF signal ($f_{RF} = 1.5$ GHz, $P_{RF} = 9$ dBm) and LO signal ($f_{LO} = 1.4$ GHz, $P_{LO} = 9$

dBm) are power combined and fed to the gate input of the mixer and the output spectra is measured with a signal analyzer, shown in Fig. 5e, where the intermediate frequency (100 MHz) along with all expected harmonics is clearly shown. This result represents the first demonstration of gigahertz MoS$_2$ mixer on flexible substrates showing great potential of bilayer MoS$_2$ for flexible RF communication. As shown in Fig. 5f, the conversion gain increases monotonically as the LO power increases from 3 to 9 dBm, similar to the rigid substrate case. The IF gains at various frequencies can be found in Supplementary Note 18. By further improving the DC performance and employing impedance matching techniques, the conversion gain can be further improved to match those on high resistivity rigid substrates[49,50].

## Discussion

Systematic study on the large area synthesis of single-crystal bilayer MoS$_2$ films on molten glass using chemical vapor deposition has been carried out. The largest domain size achieved is up to 200 μm with optimized growth condition. The transistors fabricated based on bilayer MoS$_2$ show a high field-effect mobility as well as high ON-current. Notably, the ON-current reaches a record high value at 4.3 K on a short channel 40 nm device, among the highest in 2D materials. Moreover, high performance radio frequency transistors based on these bilayer MoS$_2$ are successfully demonstrated with record high extrinsic $f_T$ and $f_{max}$ based on top-gated RF transistors. Furthermore, frequency mixers

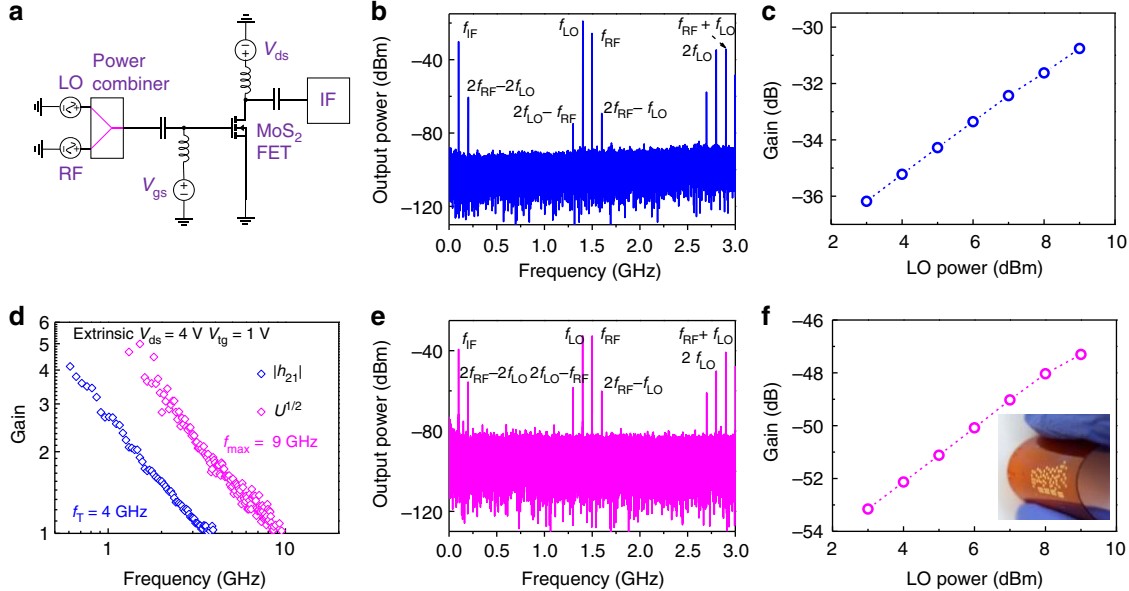

**Fig. 5** Gigahertz MoS$_2$ frequency mixer on rigid and flexible substrates. **a** Circuit schematic of MoS$_2$ FET-based RF mixer. **b** Output frequency spectrum of the mixer on rigid substrates with $f_{RF} = 1.5$ GHz, $P_{RF} = 2$ dBm and $f_{LO} = 1.4$ GHz, $P_{LO} = 9$ dBm. **c** Conversion gain as a function of LO power for the demonstrated mixer on rigid substrates. $f_{RF} = 1.5$ GHz, $P_{RF} = 2$ dBm, and $f_{LO} = 1.4$ GHz. **d** Small-signal current gain $|h_{21}|$ (blue open diamonds) and unilateral power gain $U$ (magenta open diamonds) versus frequency for device on polyimide with gate length of 300 nm. The extrinsic $f_T$ and $f_{max}$ are 4 and 9 GHz, respectively. **e** Output frequency spectrum of the mixer on flexible substrates. $f_{RF} = 1.5$ GHz, $P_{RF} = 9$ dBm and $f_{LO} = 1.4$ GHz, $P_{LO} = 9$ dBm. **f** Conversion gain as a function of LO power for the demonstrated mixer on flexible substrates. $f_{RF} = 1.5$ GHz, $P_{RF} = 9$ dBm, and $f_{LO} = 1.4$ GHz. Inset shows a photograph of MoS$_2$ circuits on flexible substrates

operating at gigahertz regime are demonstrated on rigid and flexible substrates for the first time. This work demonstrates the potential of CVD bilayer MoS$_2$ for high frequency applications and flexible wireless communication.

## Methods

**Bilayer MoS$_2$ growth and characterization.** The bilayer MoS$_2$ films were grown on molten soda-lime-silica glass substrates by atmospheric pressure CVD. Prior to growth, the substrates were cleaned in acetone, isopropyl alcohol, and deionized water, followed by 5 min of O$_2$ plasma treatment. Before the rise of temperature, the tube was pumped down to a base pressure, and followed by filling the tube with Ar to 1 atm pressure. Then, the temperatures of the zones I and II were raised to 230 °C and 830 °C, respectively. In the growth stage, 40 sccm Ar was used as carrier gas. The sulfur precursor (1.4 g) was loaded in an alumina boat and placed in zone I. The sulfur weight is adequate, determined by the experiment results of different sulfur weight. The MoO$_3$ precursor was loaded in a SiO$_2$/Si substrate and placed in zone II. The molten glass was loaded in a piece of Mo foil, which was located on the surface of a quartz plate, placed in Zone II and next to the MoO$_3$ plate. The growth durations for all samples in this work were kept as 10 min The morphology and structure of the bilayer MoS$_2$ were characterized with optical microscopy, AFM (Shimadzu SPM-9700), Raman spectroscopy (LabRAM HR800, 532 nm laser wavelength) and HRTEM (Titan G2 60-300, at 300 kV).

**Device fabrication.** Back-gated devices are fabricated on HfLaO dielectrics on highly degenerated silicon substrates, where the high-$\kappa$ dielectrics layer was deposited by atomic layer deposition (ALD). Bilayer MoS$_2$ was patterned with an electron beam lithography (EBL) step and etched using O$_2$/Ar plasma. Source and drain electrodes were formed with 20 nm Ni/60 nm Au metal stack.

Top-gated RF devices are fabricated on both silicon and polyimide substrates. Bilayer MoS$_2$ domains were transferred onto highly resistive HfLaO/Si or polyimide substrates and patterned with an EBL step, and etched using O$_2$/Ar plasma. Source and drain electrodes were formed with 20 nm Ni/60 nm Au metals stack. A thin layer of naturally oxidized Al$_2$O$_3$ and an additional layer of HfO$_2$ grown by ALD formed the top-gated dielectrics. The thickness of naturally oxidized Al$_2$O$_3$ and ALD-grown HfO$_2$ layer are 6 nm and 11 nm, respectively. The overall gate capacitance is 0.36 μF cm$^{-2}$. Two-fingered top-gates (20 nm Ni/60 nm Au) were defined by a final EBL and lift-off process.

**Device measurement.** The DC transport measurements were carried out using a Lakeshore probe station and an Agilent B1500A semiconductor parameter analyzer

with an Agilent vector network analyzer (N5225A) for high frequency measurement. The on-chip microwave measurements are carried out in the range of 10 MHz–30 GHz. Before the microwave measurements, Short-Open-Load-Thru calibrations are done with standard calibration substrates (GGB CS-5). The mixer measurements are carried out in Lakeshore probe station at room temperature using an Agilent 5182B (or Agilent N5224A) signal generator and Ceyear AV1464B signal generator as the RF and LO input source, and an Agilent DSA90804A digital (or Agilent N9030B signal analyzer) for the IF signal detection. Bias-Tee (Keysight 11612B) are used both at the input and the output to combine DC and RF signals, and provide isolation between them. The LO and RF inputs were combined using external power combiner (Keysight 11636C). Coaxial cable with SMA connectors (Rosenberger LA3-C138-100, Rosenberger LU8-C043-1500, SUCOFLEX 101PEA) were used for the signal transmission and the IF signal detection between output bias-tee and signal analyzer. All the instruments, cables, and connectors met the frequency requirements for the mixer measurement. It should be noted that none of the impedance matching techniques were used in this work. Our measurements were carried out in vacuum to avoid the effects of adsorbents from measurement environment.

## Data availability

The data that support the findings within this study are available from the corresponding author upon reasonable request.

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

## Acknowledgements

This work was supported by the National Natural Science Foundation of China (Grant Nos. 61574066, 61390504, and 61874162). Here, we wish to thank the technical support from the Center of Micro-Fabrication and Characterization, WNLO, and the Analytical and Testing Center of Huazhong University of Science and Technology. We thank J. Su from WNLO of Huazhong University of Science and Technology for TEM and SEM technique support, and J. Chen and M. Wang for the valuable discussions on $MoS_2$ synthesis and fabrication and measurement, respectively.

## Author contributions

Y.W. conceived the idea and supervised the research. Y.W., Q.G., and Z.Z. designed the experiments. Z.Z., X.X., J.S., and Q.G. performed CVD $MoS_2$ growth and analysis. Z.Z. and Q.G. performed AFM, Raman characterization and analysis, and TEM characterization and analysis. Z.Z., Q.G., and J.S. performed back-gated device fabrication, measurement, and data analysis. Q.G. performed high frequency device and flexible device fabrication, measurement, and analysis. X.L. contributed on the high-κ dielectrics growth. Y.W., Q.G., and Z.Z. co-wrote the manuscript and all authors discussed the results and contributed to the manuscript.
