## [Peer Review File · Nature Communications]

Reviewers' comments:

Reviewer #1 (Remarks to the Author):

- The linear relationship between the weight of MoO₃ and growth rate is a well-established one. Need to explain the driving mechanism to increase the domain size? What is the exact weight for MoO₃ to obtain 200 μm? If authors increased the MoO₃ weight beyond 6mg, what happened? Why authors did not alter the sulfur weight and how it was fixed at 1.4g. Have authors used constant deposition time for all the samples?
- Need to provide the other sizes of domains with different weight of MoO₃ in supporting.
- Should be include the detailed experiment part to transfer the bilayer MoS₂ on HfLaOx/Si or polyimide. And have to elaborate the merits compared with the mechanically exfoliation?
- There is no clear representation for boundary in the Fig. 2f, explain?
- Authors have demonstrated mono- and bi-layer in a single AFM image (Fig. 1f). That mean there is no uniform growth of MoS₂. Should be explain?
- There is lack of scientific discussion to obtain high mobility and on-current at 4.3 K.
- What is the role of molten glass substrate?
- Authors have highlighted triangle shape in Fig. 2h. But there is no AA structure. Also, need to provide the more TEM surface images in order to confirm the bilayer structure.
- Should be include the SEM images.
- In the Supplementary Table 1, authors should be present the values clearly.
- The provided line graphs in the Figure 4-6 are poor quality, need to improve.

Reviewer #2 (Remarks to the Author):

Authors have done a good work on synthesis of large domain MoS₂ and circuits using it. I recommend publication of this manuscript after authors have answered these questions.

1. I want to see a comparison plot of this work against those published in literature. I see many papers on synthesis of large area MoS₂, flexible MoS₂ transistors and circuits, graphene transistors and circuits... This table can compare size of film, mobility, RF performance- f_t , f_{max} ; circuit performance,...
2. Can you explain in detail why there is significant reduction in f_t / f_{max} for devices fabrication on flexible substrate - Fig5
3. I want to see flexible amplifier measurement results along with mixer. It should be straight forward measurement. This will be interesting to circuit community, given that they have very high f_{max} .
4. I want authors to add more explanation on how to improve domain size, f_t / f_{max} ; and potential future applications.

Reviewer #3 (Remarks to the Author):

In this study the authors grew and characterized CVD large area bilayer MoS₂. The material growth methods were discussed and the material characterization revealed high quality large area domains. Using this grown bilayer CVD MoS₂ for back-gated FETs on high-k rigid substrates, the authors performed DC characterization yielding promising mobility, transconductance, and current density. The authors fabricated short channel RF FETs using bilayer CVD MoS₂ and achieved record extrinsic RF performance on both rigid and flexible substrates. The authors also tested the RF devices as a mixer, achieving GHz-regime operation.

General notes to the author: There are many grammatical and sentence structure errors in this manuscript. I have taken the liberty of making editorial corrections while going through the manuscript. It is up to you whether to accept the edits or not, however the manuscript must be thoroughly edited for subsequent submissions. Also, it is highly suggested to the authors to further improve the overall quality of figures. For example the y-axis axis title on Fig. 4f is practically unreadable. It seems the image formats are compressed as they appear grainy. The situation is even worse in the pdf format.

1. In the abstract the maximum ON current is reported for CVD bilayer MoS₂ FETs at 4.3 K. While this number will be the largest measured current drive value since it is taken at a low temperature, it does not give a fair representation of current density value, both in terms of practical operating conditions and as a comparison to other MoS₂ FETs' reported current density values. Report the highest room temperature value of current density in the abstract and conclusion. This low temperature measured current value should still be reported and discussed in the main text.

a. Low temperature measurements were taken and the mobility temperature dependence was shown in Fig. 3f. Calculate this temperature dependence as $\mu \sim T^{-\gamma}$ (determine the γ value) in the high T regime ($T > 100$ K) and based on the result determine the dominant scattering mechanism. Confirm with other CVD bilayer MoS₂ reports if applicable.

b. Clarify how the intrinsic mobility is extracted (via 4 pt. measurements, or model removing contact resistance).

2. The record f_{max} of 23 GHz is stated in the abstract. The record f_T of 7.2 GHz should also be stated in the abstract as a main figure of merit for RF transistors.

3. Regarding the comparison of transfer characteristics of back-gated monolayer and bilayer FETs... It is mentioned that that the channel length is the same in both FETs. Was everything else kept the same as well? Meaning was the channel width, oxide thickness, metals, etc. the same? Ideally the devices should have been made on the same substrate in the same fabrication batch for a true fair comparison. This removes processing-induced differences between the two cases.

4. The DC characterization was done on devices fabricated on a HfLaOx dielectric substrate.

a. If not indicated in the main text can the authors comment on how this dielectric was grown and deposited? How was the lanthanum introduced into the hafnium oxide?

b. The "x" subscript indicates the oxygen content is unknown. Can the authors perform XPS or an equivalent to determine the stoichiometry of the oxygen content?

c. Additionally, can the HfLaOx dielectric be characterized in terms of capacitance to yield a dielectric constant?

d. It has been shown in previous reports that oxygen deficient dielectrics can contribute surface charge transfer doping to MoS₂. This is a potential performance boost for the FETs in this study. To show the effect of this doping source, can the authors show the I_{ds} - V_{gs} and I_{ds} - V_{ds} data for a bilayer MoS₂ FET on HfLaOx compared with conventional SiO₂. Again the experimental parameters should be kept the same for a fair comparison?

i. Regarding the high-k dielectric doping of MoS₂, here are a few references which you may cite:

1. <https://pubs.acs.org/doi/abs/10.1021/acs.nanolett.5b00314>

2. <https://ieeexplore.ieee.org/abstract/document/7175626/>

3. <http://iopscience.iop.org/article/10.1088/2053-1583/2/4/045009/me ta>

4. <https://ieeexplore.ieee.org/abstract/document/7999404/>

5. <https://ieeexplore.ieee.org/abstract/document/7999392/>

e. Show and compare the Raman and PL data of the bilayer CVD MoS₂ as-grown on molten glass (if this is feasible), transferred onto Si/SiO₂, HfLaOx, and flexible substrates. Analyze and comment on this data. For example there should be a shift in the E_{2g} mode between the as-grown and transferred material, due to stress relaxation. Additionally check for an A_{1g} broadening and shift on the HfLaOx substrate which indicates oxygen-deficiency-induced doping of the MoS₂.

5. The gate dielectric used for the RF devices was a naturally oxidized Al₂O₃ layer with an ALD-grown HfO₂ layer. Mention the thicknesses of each layer and state the overall gate capacitance value for the RF devices. Comment on why an Al₂O₃ seed e-beam deposited layer was used in conjunction with HfO₂, as opposed to for example purely HfO₂.

6. In the RF device section it is mentioned that Fig.4b shows SEM images “exhibiting the precise alignment of gate structure to the source/drain area.” I take this to mean the gate is neither underlapped nor overlapped with respect to the source/drain contacts. If this is the case can the authors rephrase this in terms of under/overlapping and discuss the benefits of this configuration (i.e. overlapping increases C_{gs} and C_{gd} capacitance while underlapping increases series resistance).

7. The DC and RF measurements were taken using a Lakeshore probe station. Were the measurements taken in the ambient environment or under vacuum? It is well known that under vacuum the electrical characteristics are improved due to the removal of adsorbents on the device surface. The authors should make this clarification in the text.

8. If not included, calculate and report the typical drain conductance value, g_{ds}, of the bilayer MoS₂ FETs in the region of operation for RF measurement. Also calculate the output resistance, r_o.

9. When reporting the f_T and f_{max} values, indicate the DC bias point for the measurement. This gives the reader an idea of operating conditions for the measurement. Include this information in the RF plots as well.

10. Is there any reason the voltage gain, A_v, plot vs. frequency is not included? This should be readily available from the s-parameter data extracted as Z₂₁/Z₁₁. Generate and include this plot in Fig. 4.

11. Can the authors also include an I_{ds}-V_{gs} and I_{ds}-V_{ds} plot for bilayer MoS₂ on flexible substrates, either in the main or supplemental text?

12. Can the authors perform DC and RF measurements of the flexible bilayer MoS₂ devices under various bending conditions? This is the main advantage of using flexible substrates so it is instructional to know if the MoS₂ material can operate under bending conditions.

13. The study reports on the measured extrinsic values for f_T and f_{max}. Why was the de-embedding procedure performed in the other 2D RF papers not done in this study? The de-embedded f_T and f_{max} parameters give an idea of the intrinsic material quality, as well as the parasitic effects of the device geometry and composition. It is highly suggested to the authors that the de-embedded values, at least for the best device data, is determined.

a. With the intrinsic f_T value, you can calculate the material velocity saturation value, v_{sat} = 2π·f_{T,int}·L. This is a useful metric for comparing RF FETs as it takes into account gate length. Calculate this value for the best intrinsic f_T data achieved, and compare with the theoretical v_{sat} value for bilayer MoS₂. Include this data column in the supplemental Table 2 of reported results.

14. In Fig. 5b,e label the RF, LO, IF, and other intermodulated harmonic signals.

15. In Fig 5c the mixer shows linear IF gain with LO power from 3 to 8 dBm.

a. Mention this linearity range in the text.

b. The mixer on flexible substrates in Fig. 5f shows linearity from the full measured range of 3 to 9 dBm. Comment on why the flexible mixer shows better linearity.

c. Also in Fig. 5f, if possible measure higher LO power IF gain data points for the flexible device to see when it becomes non-linear.

16. Can the authors include a plot of IF conversion gain vs. frequency? At f_{RF} of 1.5 GHz the mixer shows conversion loss of 33.6 dB. It would be interesting to see at what f_{RF} the IF conversion gain is unity. Keep the input signal powers and the f_{IF} of 100 MHz the same. Repeat for the flexible mixer.

17. Please provide a few more details on the test measurement setup for the mixer. Was the test system fully coaxial with SMA connectors? Were there any frequency limitations in the test setup which were overcome? Were impedance matching techniques used?

**Point-by-point response to reviewers' comments for
manuscript NCOMMS-18-09196**

===== Authors' responses to the reviewer's comments =====

Reviewer #1 (Remarks to the Author):

1. The linear relationship between the weight of MoO₃ and growth rate is a well-established one. Need to explain the driving mechanism to increase the domain size? What is the exact weight for MoO₃ to obtain 200 μm? If authors increased the MoO₃ weight beyond 6 mg, what happened? Why authors did not alter the sulfur weight and how it was fixed at 1.4 g. Have authors used constant deposition time for all the samples?

We wish to thank the reviewers for spending valuable time to read our manuscript with insightful comments.

The linear relationship between the weight of MoO₃ and growth rate were established according to our experimental result. To explain the reaction mechanism of domain size increased with MoO₃ weight increasing, the mass-driving kinetic model of CVD MoS₂ growth was constructed as Fig. R1, referring to the graphene and TMDCs growth model by Bhaviripudi et al. and Zhou et al., respectively [1, 2]. Fig. R1a presents a schematic diagram of the kinetic model of MoS₂ film

synthesis by CVD. A boundary layer is defined as a steady state between the bulk gas flow region and substrate surface region, in which the gas flow is stagnant. There is a general chemical reaction equation for CVD:

First, the vaporized precursors $MoO_3 (g)$ and $S (g)$ are transported by the carrier gas, or diffuse along the concentration gradient on top of the substrate, and then diffuse through the boundary layer and get adsorbed on the surface. Vaporized $MoO_3 (g)$ and $S (g)$ react with each other and form the product MoS_2 at 1103 K, while another product $SO_2 (g)$ diffuses away from the surface through the boundary layer and is expelled away by the bulk gas flow. The reaction process described above can be classified into two regimes: the mass transport region controlled by the diffusion rate of precursor through the boundary layer as well as the diffusion out through this layer of the gaseous products [3], and the surface reaction region depending mainly on the substrate temperature. The equations for these two fluxes are given by:

$$F_{\text{mass-transport}} = h_g (C_g - C_s) \quad (2)$$

$$F_{\text{surface-reaction}} = K_s C_s$$

(3)

Where the $F_{\text{mass-transport}}$ is the flux of the active species through the boundary layer, $F_{\text{surface-reaction}}$ is the flux of consumed active species at the surface, h_g is the mass transport coefficient, K_s is the surface reaction constant, C_g is the concentration of gas in the bulk, and C_s is the concentration of active species at the surface. These two fluxes are successive where the slower process is the rate-limiting step during the MoS_2 synthesis. At steady state, $F_{\text{mass-transport}} = F_{\text{surface-reaction}} = F_{\text{total-flux}}$, and after eliminating C_s , $F_{\text{total-flux}}$ can be rewritten as:

$$[K_s h_g / (K_s + h_g)] C_g \quad (4)$$

Mathematically, three regimes appear: $h_g \gg K_s$ (surface reaction controlled region), $h_g \sim K_s$ (mixed region), and $h_g \ll K_s$ (mass transport limited region). At high temperatures, under typical APCVD conditions, mass transport through the boundary layer is rate limiting ($h_g \ll K_s$).

We supply much larger weight of sulfur and thus assume the sulfur vapor is sufficient for the reaction. So only the $MoO_3 (g)$ concentration is discussed in our mass-driving kinetic model. We discuss four regimes here experimentally, distinguished by the increase of weight of MoO_3 powder. In Fig. R1a, ~ 1 mg MoO_3 exhibit a monolayer single domain growth. We classify this self-limiting

growth as the low mass flux. When the MoO₃ weight increases above ~1.5 mg shown in Fig. R1b, the self-limiting growth breaks down and MoS₂ bilayer domains appear. We classify the breakdown of self-limiting growth as the high mass flux, which can provide enough source for large size nuclei for the layer-by-layer growth [4]. At high reaction temperature (1103 K), the surface reaction rate takes place much faster according to the Arrhenius term, compared with the mass transport rate ($h_g \ll K_s$). Located in the mass transport limited region, higher precursor mass flux can contribute to higher diffusion rate through boundary layer, resulting in overcoming the mass transport limit and promoting single domain growth. When the weight of MoO₃ further increases to 6 mg in Fig. R1c, more precursor diffuses through the boundary layer and react on the surface ($h_g \sim K_s$). In this mixed region, domain growth is no longer restricted by either single coefficient, and the largest single domain can be obtained. When the weight of MoO₃ further increases to 8 mg in Fig. R1d, the single domain size of bilayer decreased and thicker particulates appear. We attribute the decreasing domain size of bilayer to the excess mass flux diffuses into the boundary layer and hinders the gas product (SO₂) escaping from the boundary layer. Moreover, the excess mass flux can likely lead to gas phase reactions in the bulk gas flow, resulting in the MoS₂ particulates depositing on the substrate. The domain size dependence on MoO₃ weight is depicted in Fig. R2. The above discussion have been added in the revised Supplementary Note 1.

Fig. R1. Mass-driving kinetic model. a. Low mass flux region ($\text{MoO}_3 \sim 1 \text{ mg}$). High mass flux ($\text{MoO}_3 > 1 \text{ mg}$) at b. mass transport limited region, c. mixed region and d. mass transport limited region again.

Thanks the reviewer for point this out, the exact weight for MoO_3 to obtain $200 \mu\text{m}$ is 6 mg . And we clear the exact weight in the revised manuscript.

Fig. R2. The curve of relationship between domain sizes with MoO_3 weight

We fixed the sulfur weight because it is sufficient for the reaction, even for the 6 mg MoO_3 weight, according to the equilateral triangle shape, rather than hexagonal or concave triangle[5]. We have conducted experiments with sulfur weight less than 1.4 g , such as 0.7 or 1 g , however, MoO_3 was insufficiently sulfurization and by-product MoO_xS_y was deposited on the substrate. Moreover, we have listed several recipes from other groups as depicted in Table R1, and mono- or bi-layer MoS_2 synthesis of sulfur powder weight ranging from $7\text{-}1000 \text{ mg}$ is typically used, which is less than the weight of 1.4 g in our experiments. And, we fixed the sulfur powder weight at 1.4 g through weighing fresh sulfur powder in fresh alumina boat. We have added the explanation in the method of revised manuscript.

All of the samples were synthesized for a constant deposition time of 10 min . We have clarified this in the method of revised manuscript.

Table R1. The CVD process parameters compared with other groups.

	S powder (mg)	MoO_3 (mg)	Pressure	Quartz tube diameter (mm)	Layer	Domain size (μm)
Ref [6]	350	5	AP	32	Monolayer	20

Ref [7]	800	1	AP	51	Monolayer	305
Ref [8]	50	0.01	AP	50	Monolayer	500
Ref [9]	200	1	AP	51	Monolayer	350
Ref [10]	7	20	AP	25	Bilayer	10
Ref [11]	1000	1	AP		Bilayer	60
This work	1400	6	AP	76	Bilayer	200

2. Need to provide the other sizes of domains with different weight of MoO₃ in supporting.

Fig. R3. *Optical images of varied MoO_3 weight.*

We thank the reviewer for the advice. The 10 images in Fig. R3 exhibit the other sizes of domains with the MoO_3 weight of 1, 1.5, 3, and 6 mg, including 2 pictures per weight. Monolayer MoS_2 single-crystal size up to $\sim 50 - 82 \mu\text{m}$ can be observed when the MoO_3 weight equal to 1 mg. With the MoO_3 weight increasing to 1.5 and 3 mg, bilayer MoS_2 appear with single domain size up to $44 - 53 \mu\text{m}$ and $61 - 90 \mu\text{m}$, respectively. Further increase the weight of MoO_3 powder to 6 mg, the

bilayer domain size significantly increase to 125 - 200 μm . When the MoO_3 weight exceed 6 mg, the bilayer single-crystal size begins to decrease and thicker nucleation sites appear. We have added the domains with different weight of MoO_3 in the revised Supplementary Fig. 3.

3. Should be include the detailed experiment part to transfer the bilayer MoS_2 on HfLaO_x/Si or polyimide. And have to elaborate the merits compared with the mechanically exfoliation?

Bilayer MoS_2 was transferred on HfLaO/Si or polyimide substrate via PMMA-assisted transfer strategies. Different from conventional acidic or alkaline solutions as etchant, deionized water was utilized here for the hydrophobicity/hydrophilicity property of the as-grown $\text{MoS}_2/\text{glass}$ stack. The modified PMMA-assisted transfer process is as follows: Firstly, PMMA A4 was spin-coated on fresh $\text{MoS}_2/\text{glass}$ stack at a speed of 500 r/min for 60 s, followed by 120 $^\circ\text{C}$ bake for 5 min. Secondly, the PMMA/ $\text{MoS}_2/\text{glass}$ stack was inclined gently inside a beaker with fresh deionized water. In this way, the PMMA/ MoS_2 film can be delaminated from the glass substrate with the water penetrated into the interface between MoS_2 film and glass. Thirdly, cleaned HfLaO/Si or PI substrate was utilized here for the supporting substrate for the floating PMMA/ MoS_2 films, followed by nitrogen blow dry and hot plate baking for stronger adhesion. Then, PMMA/ $\text{MoS}_2/\text{HfLaO}/\text{Si}$ or PI stack was immersed into acetone to remove PMMA, followed by IPA wash and nitrogen blow dry. Moreover, for the sample on rigid substrate, 350 $^\circ\text{C}$ annealing in argon atmosphere for 5 hours was carried out to remove organic residual. We have added the transfer method in the revised Supplementary Note 5.

Fig. R4. *Transfer process*

Although mechanically exfoliation strategy is accessible and easy to obtain large amount MoS₂ flake on target substrate, but the uncontrollable layer thickness, random location distribution and domain size hinder the device uniformity and future large area applications [12, 13]. On the other hand, CVD growth method possesses the advantages of layer number control [14], larger domain[15]and wafer-scale growth[16, 17] by adjusting substrate geometry, growth temperature, pressure, carrier gas and etc., overcoming the obstacle of mechanical exfoliation confronted. Moreover, with the optimized synthesis, transfer and device fabrication, CVD growth monolayer or bilayer MoS₂ have higher potential of obtaining higher mobility and device performance than mechanical exfoliation ones [6, 18]. We have added the merits comparison with the mechanically exfoliation in the revised Supplementary Note 5.

4. There is no clear representation for boundary in the Fig. 2f, explain?

We appreciate the reviewer’s comment. We attribute this blurry boundary in the Fig.2f to the weak contrast in the bright field mode under 300 kV accelerating voltage. In order to clarify the issue, we performed more TEM measurement and hereby attach 3 images of few layer MoS₂ boundary in dark field STEM mode in Fig. R5. Clear representation for boundary can be observed by the high contrast. We have added the TEM images with clear boundary in the revised Supplementary Fig. 5.

Fig. R5. *Low resolution dark field STEM images of few layer MoS₂ on carbon-coated micro copper grid*

5. Authors have demonstrated mono- and bi-layer in a single AFM image (Fig. 1f). That mean there is no uniform growth of MoS₂. Should be explain?

We thank the reviewer for pointing this out. Zheng et al has synthesized multilayered MoS₂ flakes via the rapid sulfurization of Mo oxides in gas phase in a confined space [18]. Similar to their work, we synthesized MoS₂ bilayer via adequate sulfurization of high concentration MoO₃ vapor at high temperature. High concentration MoO₃ vapor bring large-size nucleus, providing enough source for the layer-by-layer growth [4]. In general, the first layer have a faster growth rate than the second layer [11], and the growth time decreased from the first to second layer [18], both of them directly results in the bilayer MoS₂ flakes stacked by two concentric triangles with shrinking size as revealed by AFM image, optical image, Raman, PL spectra and TEM. Please note that Fig. 1f is located at the edge of a representative bilayer MoS₂ flake while Fig. 1g is on a natural crack in the center region of the bilayer flake. As can be seen from the detailed reply in the second comment, the bilayer flakes are typically uniform with a monolayer edge underneath. We have added the explanation in the revised manuscript.

6. There is lack of scientific discussion to obtain high mobility and on-current at 4.3 K.

We thank the reviewer for pointing this out.

The intrinsic mobility can be fitted using the following function:

$$1/\mu(T) = 1/\mu_{\text{imp}} + 1/\mu_{\text{ph}}(T) \quad (5)$$

Where μ_{imp} represents the contribution from Coulomb impurity scattering and μ_{ph} is the temperature-dependent contribution from phonon scattering. Moreover, the fitted $\mu_{\text{ph}}(T)$ is well described by a power law ($\mu_{\text{ph}} \sim T^{-\gamma}$) above 100 K. This behavior is consistent with mobility limited MoS₂ optical phonons, which is theoretically predicted to have an exponent of ~ 1.69 in monolayer [19] and ~ 2.5 in bulk [20] MoS₂ at $T > 100$ K. A temperature dependence was observed in Fig. R6, with the exponent γ equal to 1.48, consistent with literatures [21]. The bilayer mobility begins to saturate below 100 K, a temperature by which scattering from optical phonons is expected to become negligible [22] and long-range Coulomb impurity scattering becomes dominant. High- κ gate dielectric used as the top-gate dielectrics [23] or defects and interface decoration [24] can further effectively screen phonon and coulomb impurity scattering as shown in previous studies. In short, the mobility increase from 36 cm²/Vs at 300 K to 127 cm²/Vs at 4.3 K due to the reduced phonon scattering. Moreover, on-current increase with the contribution from mobility improvement at low temperature. By pushing V_{gs} to 6 V, higher electron carrier density can be obtained and results in a record high on-current at low temperature. We have added the discussion in the revised Supplementary Note 10.

Fig. R6. The extracted carrier mobility vs. temperatures.

7. What is the role of molten glass substrate?

We appreciate the reviewer's comment. Chen et al and Yang et al synthesized TMDCs large single-crystal utilized molten glass recently and discussed the mechanism [25, 26]. Their work provides a good reference for the discussion of the role of molten glass.

Molten glass (soda-lime silica) was used here for providing a "liquid-state" like surface at growth temperature 1103 K. On the one hand, different from conventional rigid substrate (thermal SiO₂/Si, sapphire etc.), the melting and regeneration of molten glass generate a smooth surface with ultralow defect density. Due to the nucleation of crystallites occurs preferentially at impurities or defect sites, nucleation density will drastically reduce with the decrease of defect sites, thus newly-generated nucleus mainly contribute to the growth of the MoS₂ films rather than forming new nucleation, which promoting the growth of large single-crystal [25].

On the other hand, molten glass lowers the migration barrier energy (U) of adatoms. The migration coefficient D is related to U by

$$D \approx D_{\infty} \exp(-U/k_B T) \quad (6)$$

Where k_B is the Boltzmann constant and T is the temperature [27]. The migration coefficient on molten glass is much higher than that on solid substrates, which contributes to a high growth rate. Moreover, Yang et al had conducted experiments and confirmed that Na element in the soda-lime glass is considered to serve as an intermediate catalyst in the rapid growth of MoS₂ [26]. Besides, because of the hydrophobicity/hydrophilicity of the MoS₂/glass substrate, MoS₂ film can be separated from glass immediately by immersing in deionized water. This short-time and safe transfer method make MoS₂ film free from structural damage and performance degradation. Furthermore, PMMA, gold or other films can be coated or deposited on MoS₂/glass stack as a supporting layer if necessary for large area transfer or contaminant-free contact. Overall, molten glass plays an important role in promoting MoS₂ large single-crystal growth and etching-free, short-time transfer. We have added the discussion in the revised Supplementary Note 2.

8. Authors have highlighted triangle shape in Fig. 2h. But there is no AA structure. Also, need to provide the more TEM surface images in order to confirm the bilayer structure.

We thank the reviewer for pointing this point. We have made a correction of Fig. 2h about atomic

configuration of AA stacked MoS₂ bilayer. In order to confirm the bilayer structure, more TEM surface images have been provided as follows. Fig. R7a-d show HRTEM surface image of AA stacked bilayer MoS₂ on carbon film TEM grid. Fast Fourier transform (FFT) taken at Fig. R7a exhibit only one set of hexagonally arranged diffraction spots (Fig. R7e), indicating the single crystalline nature of the flake over a large area and almost 0° twisted angle between first- and second-layers [18]. Combining with the optical images of concentric triangular geometry (Fig. R2), AA stacked order of these two layer is confirmed [28], where S atoms of the top layer overlapped with the hexagonal centers of the bottom layer (Fig. R7f). We have added more TEM surface images in the revised Supplementary Fig. 6.

Fig. R7. TEM characterizations of MoS₂ bilayer. a-d. HRTEM images of AA stacked bilayer MoS₂. e. FFT image of MoS₂ bilayer. f. Atomic configuration of AA stacked MoS₂ bilayer.

9. Should be include the SEM images.

Fig. R8 a-d exhibit SEM images of as-grown bilayer and monolayer MoS₂ domains on molten glass. Accelerating voltage is 2 kV. We have added the SEM images in the revised Supplementary Fig. 4.

Fig. R8. SEM images of growth bilayer MoS_2 for varied MoO_3 weight.

10. In the Supplementary Table 1, authors should be present the values clearly.

We thank the reviewer for pointing this out. We have added the accurate values of reported works in the Supplementary Table 1. The accurate values were measured and calculated according to the scale bars in the references.

11. The provided line graphs in the Figure 4-6 are poor quality, need to improve.

We thank the reviewer for pointing this out and apologize for the degraded quality of the images during file conversion. We have provided better quality of all graphs in the revised manuscript.

~~~~~  
~~~~~

Reviewer #2 (Remarks to the Author):

Authors have done a good work on synthesis of large domain MoS₂ and circuits using it. I recommend publication of this manuscript after authors have answered these questions.

We greatly appreciate the positive comments from the reviewer.

1. I want to see a comparison plot of this work against those published in literature. I see many papers on synthesis of large area MoS₂, flexible MoS₂ transistors and circuits, graphene transistors and circuits... This table can compare size of film, mobility, RF performance- f_t, f_{max} ; circuit performance,...

A comparison of large area MoS₂, flexible MoS₂ transistors and demonstrated circuits, graphene transistors and demonstrated circuits from literatures in the past three years are listed in the followed Table R2. From this table, we can see that the results show the best RF performance compared with previous similar devices based on MoS₂, and with extrinsic RF performance comparable to the best of CVD graphene.

Table R2. Comparison of high-frequency transistors and circuits based on graphene and MoS₂

Ref	Material	μ	$f_{T,ext}$	$f_{max,ext}$	Demonstrated Circuit Performance
		cm ² /Vs	GHz	GHz	
[29]	CVD graphene on SiN/Si	--	22*	20.7*	1. Frequency mixer. f_{RF} 3.5 GHz, f_{LO} 3.6 GHz, conversion gain -33 dB.
[30]	CVD graphene on HfO ₂ /Si, (domain size 2 mm)	3300	15*	20*	1. Frequency mixer. f_{RF} 4 GHz, f_{LO} 3.8 GHz, conversion gain -31 dB.
[31]	CVD graphene on SiO ₂ /Si	2286	--	--	1. Frequency doubler. f_{RF} 20 kHz, conversion gain -22 dB.
[32]	Exitaxial graphene on SiC	6500	38	27	1. MMIC mixer. f_{RF} 90 GHz, f_{LO} 89 GHz, conversion gain -18 dB.
[33]	CVD bilayer graphene on HfSiO/Si (domain size 600 μ m)	2300	13	16	1. Frequency mixer. f_{RF} 2.2 GHz, f_{LO} 2 GHz, conversion gain -7 dB.

[34]	CVD MoS ₂ on SiO ₂ /Si	55	2.8	3.6	1. Frequency amplifier, f_{RF} 1.4 MHz, gain 14 dB; 2. Frequency mixer, f_{RF} 1.4 MHz, f_{LO} 1.1 MHz, conversion gain - 15 dB.
[35]	CVD MoS ₂ on polyimide	22	2.7	2.1	1. Flexible amplifier, f_{RF} 1.4 MHz, gain 15 dB; 2. Frequency mixer, f_{RF} 1.4 MHz, f_{LO} 1.1 MHz, conversion gain - 17 dB.
This work	CVD bilayer MoS ₂ on HfLaO/Si (domain size 200 μ m)	36	7.2	23	1. Frequency mixer, f_{RF} 1.5 GHz, f_{LO} 1.4 GHz, conversion gain - 30.7 dB.
	CVD bilayer MoS ₂ on polyimide (domain size 200 μ m)		4	9	1. Flexible mixer, f_{RF} 1.5 GHz, f_{LO} 1.4 GHz, conversion gain - 47 dB.

Notes: * f_T and f_{max} after de-embedding

2. Can you explain in detail why there is significant reduction in f_i/f_{max} for devices fabrication on flexible substrate - Fig5

We thank the reviewer for pointing this out. The f_T and f_{max} for devices on flexible substrate in Fig. 5 are from a 300 nm gate length devices, which is comparable to the device performance on rigid substrate with the same gate length as shown in Fig. 4f and Fig. 4g. And both are much smaller than the best 90 nm device on rigid substrate because of the channel length scaling properties.

3. I want to see flexible amplifier measurement results along with mixer. It should be straight forward measurement. This will be interesting to circuit community, given that they have very high f_{max} .

We thank the reviewer for pointing this out. A flexible bilayer MoS₂ amplifier demonstrated in this work is shown in Fig. R9 (a). The input and output voltage waveform of the flexible amplifier at 1.2 GHz is shown in Fig. R9 (b). A gain of -37 dB at 1.2 GHz is achieved. The unsatisfactory gain is mainly due to the mismatch at the input and out port. Maximum oscillation frequency f_{max} is a simple, and highly practical, figure of merit derived from the unilateral gain U. It is defined as the frequency at which U becomes unity. The unilateral power gain U is the maximum power gain that

can be obtained from the two-port network, after it has been made unilateral with the help of a lossless and reciprocal embedding network. [36] Fig. R9(c) shows the as measured input port voltage reflection coefficient S11 and output port voltage reflection coefficient S22. Both S11 and S22 are close to 1, which show the mismatch of the transistor with the measurement system. Thus, in order to obtain an amplifier with positive gain, input and output match technique should be adapted. These techniques are including device structure optimization including much larger gate width and external match circuit design. The present work is focused on the novel material synthesis and high performance DC and RF transistors demonstrations, and further efforts on extensive circuit design and RF modeling will be carried out in future work.

Fig. R9. (a) Measurement setup for the amplifier based on CVD bilayer MoS₂ flexible transistors. (b) Measured input and output waveform of a flexible MoS₂ transistors based amplifier. The device is biased at $V_{DD} = 4$ V, $V_{gs} = 0.3$ V. The input frequency is at 1.2 GHz. (c) S11 and S22 vs. frequency for the flexible MoS₂ transistors.

4. I want authors to add more explanation on how to improve domain size, f_t/f_{max} ; and potential future applications.

Molten glass (soda-lime silica) was used here for providing a “liquid-state” like surface at growth temperature 1103 K. On the one hand, different from conventional rigid substrate (thermal SiO₂/Si,

sapphire or else), the melting and regeneration of molten glass generate a smooth surface with ultralow defect density. To improve domain size of CVD bilayer MoS₂, low nucleation density is needed [37]. Due to the nucleation of crystallites occurs preferentially at impurities or defect sites, nucleation density will drastically reduce with the decrease of defect sites, thus newly generated nucleus mainly contribute to the growth of the MoS₂ films rather than forming new nucleation, which promoting the growth of large single-crystal [25].

On the other hand, molten glass lowers the migration barrier energy (U) of adatoms. The migration coefficient D is related to U by

$$D \approx D_{\infty} \exp(-U/k_B T) \quad (6)$$

Where k_B is the Boltzmann constant and T is the temperature [27]. The migration coefficient on molten glass is far higher than that on solid substrates, which contributes to a high growth rate. Moreover, Yang et al had conducted experiments and confirmed that Na element in the soda-lime glass is considered to serve as an intermediate catalyst in the rapid growth of MoS₂ [26].

In short, low defect sites, high migration coefficient and Na boost growth are important factors in realizing large domain size. Also, with adequate amounts MoO₃ supply, bilayer MoS₂ growth can overcome the limit of mass-transport, thus large domains were obtained. We have added the discussion in the revised Supplementary Note 1 and 2.

The cut-off frequency (f_T), defined as the frequency at which the current gain becomes unity, is one of the most important figures-of-merit for evaluating the performance of RF devices. In a well-behaved field-effect transistor, the intrinsic cut-off frequency can be related to trans-conductance g_m by the following equation: $f_T = \frac{g_m}{2\pi C_g}$, where C_g is the gate capacitance.

And, actually trans-conductance g_m is associated with carrier mobility μ , gate voltage V_{gs} , threshold voltage V_{th} and gate length L_g . Thus, for the devices with long gate length, $f_T \approx \frac{\mu(V_{gs} - V_{th})}{2\pi L_g^2}$, and for

the devices with short gate length, $f_T \approx \frac{v_{sat}}{2\pi L_g}$. Also, a major limitation on the performance of

MoS₂ for RF applications arises from the high values of contact resistance R_c determining the source and drain parasitic resistances [38]. The lower contact resistance result from higher density of states and carrier mobility of CVD bilayer MoS₂ when compared with monolayer one. Thus, high carrier

mobility, low contact resistance and short gate lengths can lead to higher f_T of MoS₂ transistors. Maximum oscillation frequency, f_{\max} , which is defined as the frequency at which the power gain equals to unity. f_{\max} can be calculated as $f_{\max} = \frac{f_T}{2\sqrt{g_{ds}(R_g + R_s) + 2\pi f_T R_g C_{gd}}}$, where g_{ds} is the output conductance, R_g is the gate resistance, R_s is the source resistance, and C_{gd} is gate-to-drain capacitance [39]. As shown in Fig. R10, the typical drain-source conductance $g_{ds} = \partial I_{ds} / \partial V_{ds}$ are smaller than 10 $\mu\text{S}/\mu\text{m}$ in the region of operation for RF measurement. And, the corresponding r_o ($r_o = 1/g_{ds} = \partial V_{ds} / \partial I_{ds}$) are larger than 0.1 $\text{M}\Omega\cdot\mu\text{m}$. As a result, further improvement of f_{\max} can be obtained by higher f_T , high output resistance and reduction of gate resistance and other parasitic resistances by device structure optimizations. We have added the discussion in the revised Supplementary Note 12.

Fig. R10. Typical output conductance vs. drain voltage for the device with gate length of 90 nm (a) and 300 nm (b).

The availability of potential large area production, the ultrathin body nature as well as attractive electronic properties makes MoS₂ a promising candidate for future ubiquitous electronics. Potential future applications including thin film transistor applications on flexible substrates, ultrashort channel transistors, vertical integration on various heterostructure for tunneling devices, wearable sensing and communication devices. We have added more applications in the revised manuscript.

~~~~~  
 ~~~~~

Reviewer #3 (Remarks to the Author):

In this study the authors grew and characterized CVD large area bilayer MoS₂. The material growth methods were discussed and the material characterization revealed high quality large area domains. Using this grown bilayer CVD MoS₂ for back-gated FETs on high-k rigid substrates, the authors performed DC characterization yielding promising mobility, trans-conductance, and current density. The authors fabricated short channel RF FETs using bilayer CVD MoS₂ and achieved record extrinsic RF performance on both rigid and flexible substrates. The authors also tested the RF devices as a mixer, achieving GHz-regime operation.

General notes to the author: There are many grammatical and sentence structure errors in this manuscript. I have taken the liberty of making editorial corrections while going through the manuscript. It is up to you whether to accept the edits or not, however the manuscript must be thoroughly edited for subsequent submissions. Also, it is highly suggested to the authors to further improve the overall quality of figures. For example the y-axis axis title on Fig. 4f is practically unreadable. It seems the image formats are compressed as they appear grainy. The situation is even worse in the pdf format.

We greatly appreciate the reviewer for the corrections of grammatical and sentence structure errors. All the grammatical and sentence structure errors have been corrected in the revised manuscript according to the reviewer's suggestions. Also the quality of figures have been improved in the revised manuscript.

1. In the abstract the maximum ON current is reported for CVD bilayer MoS₂ FETs at 4.3 K. While this number will be the largest measured current drive value since it is taken at a low temperature, it does not give a fair representation

of current density value, both in terms of practical operating conditions and as a comparison to other MoS₂ FETs' reported current density values. Report the highest room temperature value of current density in the abstract and conclusion. This low temperature measured current value should still be reported and discussed in the main text.

We thank the reviewer for pointing this out. The highest room temperature value of current density has been added in the abstract and conclusion of revised manuscript.

a. Low temperature measurements were taken and the mobility temperature dependence was shown in Fig. 3f. Calculate this temperature dependence as $\mu \sim T^{-\gamma}$ (determine the γ value) in the high T regime ($T > 100$ K) and based on the result determine the dominant scattering mechanism. Confirm with other CVD bilayer MoS₂ reports if applicable.

Fig. R11. The extracted carrier mobility vs. temperatures.

The intrinsic mobility curves can be reasonably fitted to a functional form:

$$1/\mu(T) = 1/\mu_{\text{imp}} + 1/\mu_{\text{ph}}(T) \quad (5)$$

Where μ_{imp} represents the contribution from Coulomb impurity scattering and μ_{ph} is the temperature-dependent contribution due to phonon scattering. Moreover, the fitted $\mu_{\text{ph}}(T)$ is well described by a power law ($\mu_{\text{ph}} \sim T^{-\gamma}$) above 100 K. This behavior is consistent with mobility limited by MoS₂ optical phonons, which is theoretically predicted to have an exponent of ~ 1.69 in monolayer [19] and ~ 2.5 in bulk [20] MoS₂ at $T > 100$ K. A temperature dependence was

observed in Fig. R11, with the exponent γ of 1.48, consistent with other results on MoS₂ bilayer from table R3 which varies from 1.1 to 2.5 [21]. The bilayer mobility begins to saturate below 100 K, a temperature by which scattering from optical phonons is expected to become negligible [22]. Long-range Coulomb impurity scattering is dominated in this region and further experimental and theoretical study are needed. High- κ material with higher phonon energy used as the top-gate dielectrics [23] or defects and interface decoration [24] can further effectively screen phonon and Coulomb impurity scattering. We have added the discussion in the revised Supplementary Note 10.

Table R3. Comparison of γ value from other groups

	Layer number	Process	γ value	Method
Ref [21]	Bilayer	Exfoliation	2.5	hBN-encapsulation
Ref [22]	Monolayer	CVD	0.7	Two-step annealing
Ref [23]	Monolayer	Exfoliation	0.55-0.78	HfO ₂ encapsulation
Ref [24]	Monolayer	Exfoliation	0.72	DS-treated
Ref [40]	Bilayer	Exfoliation	1.1	Vacuum annealing
This work	Bilayer	CVD	1.48	HfLaO back-gate dielectric

b. Clarify how the intrinsic mobility is extracted (via 4 pt. measurements, or model removing contact resistance).

We thank the reviewer for pointing this out. We use transfer length method to remove contact resistance. Field-effect mobility is firstly extracted from the transfer characteristics in the linear region using the equation $\mu_{FE} = g_m L_{ch} / WC_{ox} V_{ds}$, where g_m is the peak transconductance, L_{ch} and W are the channel length and width, respectively, C_{ox} is the back gate capacitance, and V_{ds} is the drain-to-source voltage. Then, to eliminate the parasitic effects from contact resistance, the

effective V_{gs} , V_{ds} , and intrinsic transconductance are given by $V_{gs}' = V_{gs} - I_{ds}R_c$, $V_{ds}' = V_{ds} - 2I_{ds}R_c$, and $g_m' \approx g_m/(1 - R_c g_m)$, respectively, where R_c is the contact resistance extracted by the transfer length method [41, 42]. We have added the method of intrinsic mobility extraction in the revised Supplementary Note 8.

2. The record f_{max} of 23 GHz is stated in the abstract. The record f_T of 7.2 GHz should also be stated in the abstract as a main figure of merit for RF transistors.

We thank the reviewer for pointing this out. The record f_T of 7.2 GHz have been stated in the abstract of revised manuscript.

3. Regarding the comparison of transfer characteristics of back-gated monolayer and bilayer FETs... It is mentioned that that the channel length is the same in both FETs. Was everything else kept the same as well? Meaning was the channel width, oxide thickness, metals, etc. the same? Ideally the devices should have been made on the same substrate in the same fabrication batch for a true fair comparison. This removes processing-induced differences between the two cases.

We thank the reviewer for pointing this out. For the comparison of transfer characteristics of back-gated monolayer and bilayer FETs, the devices were made on the same substrate with the same oxide thickness and in the same fabrication batch to avoid processing-induced differences. We have clarified this in the revised manuscript.

4. The DC characterization was done on devices fabricated on an HfLaOx dielectric substrate.

a. If not indicated in the main text can the authors comment on how this dielectric was grown and deposited? How was the lanthanum introduced into the hafnium oxide?

Atomic layer deposition (ALD) was utilized here for deposition HfLaO film in this work. Before

deposition, the substrate was cleaned by standard RCA cleaning process and a diluted buffered oxide etch (BOE) dip to remove organic and metallic contaminants, particles and unintentional oxides, followed by a deionized water rinse and drying. The substrate was then transferred to an ALD chamber to deposit HfLaO film at 300 °C, using TEMA₂Hf, La((¹Pr₂N)₂CH)₃ and O₃ as the Hf precursor, La precursor and oxygen source, respectively. The HfLaO film was achieved by controlling the HfO₂:La₂O₃ cycle ratio of 8:1. The rapid thermal annealing process was then performed in nitrogen ambient for 30 s at 500 °C. We have added the method of HfLaO grown in the revised Supplementary Note 6.

b. The “x” subscript indicates the oxygen content is unknown. Can the authors perform XPS or an equivalent to determine the stoichiometry of the oxygen content?

We thank the reviewer for pointing this out. La 3d and Hf 4f XPS spectra are shown in Fig. R12, with the stoichiometric ratio of La : Hf is 0.06 : 1. According to the XPS analysis, La:HfO_x can be represented as La:HfO_{1.95}, with 0.14 portion of oxygen deficiency. We have added the XPS result of HfLaO in the revised Supplementary Fig. 10.

Fig. R12. XPS spectra of the (a) La 3d, (b) Hf 4f in the La:HfO_x film..

c. Additionally, can the HfLaO_x dielectric be characterized in terms of capacitance to yield a dielectric constant?

70 nm hafnium lanthanum oxide (HfLaO) was deposited as dielectric insulator on doped silicon substrates by ALD for MOSCAP fabrication. C - V measurement was performed at varied frequency from 50 - 1000 kHz, and oxide capacitance C_{ox} was extracted at 1000 kHz equal to $0.34 \mu\text{F}/\text{cm}^2$, as shown in Fig. R13. According to the formula of parallel plate capacitance, relative dielectric constant $\kappa = \epsilon/\epsilon_0 = C_{ox} t_{ox} / \epsilon_0 = 26.7$, where ϵ represents dielectric constant, ϵ_0 represents vacuum dielectric constant, and t_{ox} represents oxide thickness. We have added the CV result of HfLaO in the revised Supplementary Fig. 9.

Fig. R13. MOSCAP C - V measurement results.

d. It has been shown in previous reports that oxygen deficient dielectrics can contribute surface charge transfer doping to MoS_2 . This is a potential performance boost for the FETs in this study. To show the effect of this doping source, can the authors show the I_{ds} - V_{gs} and I_{ds} - V_{ds} data for a bilayer MoS_2 FET on HfLaOx compared with conventional SiO_2 . Again the experimental parameters should be kept the same for a fair comparison?

Our response: We thank the reviewer for pointing this out. Bilayer MoS_2 devices on 90 nm SiO_2/Si and 70 nm HfLaO/Si were fabricated with the same experimental process side by side. Equivalent oxide thickness (EOT) of 70 nm HfLaO equals to 10.5 nm. Because of the EOT difference of 8.57 times, the V_{gs} in the I_{ds} - V_{gs} plot of the HfLaO device is multiplied by the same factor for a fair comparison with the 90 nm SiO_2 device, which is $V_{gs}^* = V_{gs} \cdot \frac{90}{\text{EOT}}$, where V_{gs} remains unchanged for

the 90 nm SiO₂ substrate, but multiplies 8.57 for HfLaO/Si substrate, respectively as depicted in Fig. R14a. We can find a negative V_T shift about -12.4 V on HfLaO/Si substrate, compared with SiO₂/Si, indicating a charge transfer and positive fixed charge [43]. Compared with 0.44 portion oxygen deficiency in HfO_x [44], our oxygen deficiency of 0.14 in La:HfO_x induce a moderate V_T shift than in literature, indicating an interfacial oxygen vacancy doping as in many literatures. MoS₂ on HfLaO/Si substrate exhibit a higher ON current and lower contact resistance than on SiO₂/Si, as depicted in the I_{ds} - V_{ds} plot in Fig. R14b. We have added the DC result of bilayer MoS₂ on HfLaO/Si and SiO₂/Si substrates in the revised Supplementary Fig. 12.

Fig. R14. *a. Transfer and b. Output characteristics curves of MoS₂ bilayer device on SiO₂/Si and HfLaO/Si substrate.*

i. Regarding the high-k dielectric doping of MoS₂, here are a few references which you may cite:

1. <https://pubs.acs.org/doi/abs/10.1021/acs.nanolett.5b00314>
2. <https://ieeexplore.ieee.org/abstract/document/7175626/>
3. <http://iopscience.iop.org/article/10.1088/2053-1583/2/4/045009/meta>
4. <https://ieeexplore.ieee.org/abstract/document/7999404/>
5. <https://ieeexplore.ieee.org/abstract/document/7999392/>

We thank the reviewer for providing the useful references. We have cited the above references in the revised manuscript as ref. 38-42.

e. Show and compare the Raman and PL data of the bilayer CVD MoS₂ as-grown on molten glass (if this is feasible), transferred onto Si/SiO₂, HfLaOx, and flexible substrates. Analyze and comment on this data. For example there should be a shift in the E_{2g} mode between the as-grown and transferred material, due to stress relaxation. Additionally check for an A_{1g} broadening and shift on the HfLaOx substrate which indicates oxygen-deficiency-induced doping of the MoS₂.

We thank the reviewer for detailed and constructive comments. Raman and PL spectrum of the bilayer CVD MoS₂ have been performed on three different conditions: as-grown on molten glass, transferred onto SiO₂/Si, and HfLaO/Si substrate, as depicted in Fig. R15. Raman and PL spectrum of the bilayer CVD MoS₂ transferred onto polyimides (PI) substrate was performed as well, but polymeric substances have highly variable physical structures that give a wide range of Raman patterns, overwhelming the CVD bilayer MoS₂ Raman signal [45] and is thus not included in this comparison. There are two shifts in the E_{2g} and A_{1g} mode between the as-grown and transferred CVD MoS₂ bilayer, due to the strain stress release [46]. Moreover, A_{1g} peak redshift and FWHM broadening in Raman and A1 exciton redshift in PL spectrum are observed on HfLaO/Si substrate, indicating an interfacial oxygen vacancy doping of the MoS₂ [71]. We have added the Raman and PL results in the revised Supplementary Fig. 11.

Fig. R15. *a.* Raman and *b.* PL spectrum of CVD MoS₂ bilayer as-grown on molten glass, transferred onto SiO₂/Si and HfLaO/Si substrates

5. The gate dielectric used for the RF devices was a naturally oxidized Al_2O_3 layer with an ALD-grown HfO_2 layer. Mention the thicknesses of each layer and state the overall gate capacitance value for the RF devices. Comment on why an Al_2O_3 seed e-beam deposited layer was used in conjunction with HfO_2 , as opposed to for example purely HfO_2 .

We thank the reviewer for pointing this out. Following the suggestion, we have added the thickness of each layer and state the overall gate capacitance value for the RF devices in the method of revised manuscript. The thickness of naturally oxidized Al_2O_3 and ALD-grown HfO_2 layer are 6 nm and 11 nm, respectively, which were determined through AFM measurement, as shown in Fig. R16. The overall gate capacitance is $0.36 \mu\text{F}/\text{cm}^2$. Although researchers have reported the formation of a uniform HfO_2 layer by ALD in part of the MoS_2 transistor fabrication. The Wallace group proved that HfO_2 was not continuous on the surface of MoS_2 flakes by results and DFT calculations and claimed that the uniform growth in previous reports was probably due to the existence of adhesive layer residues that promoted ALD nucleation [47, 48]. Thus, an Al_2O_3 seed e-beam deposited layer were used to ensure uniform deposition of thin HfO_2 in this work. Future improvement on RF transistors directly using HfO_2 or HfLaO while avoiding the uniformity concern can be achieved using embedded gate structures as in our other work on black phosphorus RF transistor [49]. (Adv. Electron. Mater. 2018, 4, 1800138).

Fig. R16 (a) AFM microscope of naturally oxidized Al_2O_3 . Inset shows the naturally oxidized Al_2O_3

with thickness around 6 nm. (b) AFM microscope of naturally oxidized Al_2O_3 and ALD-grown HfO_2 . Inset shows the thickness around 17 nm ($Al_2O_3 + HfO_2$).

6. In the RF device section it is mentioned that Fig.4b shows SEM images “exhibiting the precise alignment of gate structure to the source/drain area.” I take this to mean the gate is neither underlapped nor overlapped with respect to the source/drain contacts. If this is the case can the authors rephrase this in terms of under/overlapping and discuss the benefits of this configuration (i.e. overlapping increases C_{gs} and C_{gd} capacitance while underlapping increases series resistance).

We thank the reviewer for pointing this out. In this manuscript, the precise alignment of gate structure means the lengths of gate-to-source L_{gs} and gate-to-drain L_{gd} are almost the same in such short channel length. There is no overlap in our devices design to avoiding the increases of C_{gs} and C_{gd} capacitance. At the same time, the length of L_{gd} and L_{gs} is short to decrease the series resistance. From SEM images, the underlap is also minimized using optimized E-beam lithography process to avoid increase of access resistance, which is critical in the RF performance as well. We have clarified this in the revised manuscript.

7. The DC and RF measurements were taken using a Lakeshore probe station. Were the measurements taken in the ambient environment or under vacuum? It is well known that under vacuum the electrical characteristics are improved due to the removal of adsorbents on the device surface. The authors should make this clarification in the text.

Our measurements were carried out in vacuum to avoid the effects of adsorbents from measurement environment. As the reviewer suggested, we have added the information in the methods section of revised manuscript.

8. If not included, calculate and report the typical drain conductance value, g_{ds} ,

of the bilayer MoS₂ FETs in the region of operation for RF measurement. Also calculate the output resistance, r_o .

As shown in Fig. R17, the typical drain-source conductance $g_{ds} = \partial I_{ds} / \partial V_{ds}$ are smaller than 10 $\mu\text{S}/\mu\text{m}$ in the region of operation for RF measurement. And, the corresponding r_o ($r_o = 1 / g_{ds} = \partial V_{ds} / \partial I_{ds}$) are larger than 0.1 $\text{M}\Omega \cdot \mu\text{m}$. The output conductance and output resistance have been added in the revised Supplementary Note 11 as the reviewer suggested.

Fig. R17. Typical output conductance vs. drain voltage for the device with gate length of 90 nm (a) and 300 nm (b).

9. When reporting the f_T and f_{\max} values, indicate the DC bias point for the measurement. This gives the reader an idea of operating conditions for the measurement. Include this information in the RF plots as well.

We thank the reviewer for detailed and constructive comments. The DC bias point for the measurement have been added in the revised manuscript. Also, this bias information have been added in the RF plots.

10. Is there any reason the voltage gain, A_v , plot vs. frequency is not included? This should be readily available from the s-parameter data extracted as Z_{21}/Z_{11} . Generate and include this plot in Fig. 4.

We thank the reviewer for pointing this out. The voltage gain vs frequency has been added as Fig 4e

in the revised manuscript.

11. Can the authors also include an I_{ds} - V_{gs} and I_{ds} - V_{ds} plot for bilayer MoS₂ on flexible substrates, either in the main or supplemental text?

The I_{ds} - V_{gs} and I_{ds} - V_{ds} plots of a device with $L_g = 600$ nm for bilayer MoS₂ on flexible substrates have been added in the Supplementary Note 16. On/off current ratio of about 10^8 and on current density of $40 \mu\text{A}/\mu\text{m}$ were achieved. For the device with $L_g = 300$ nm, On/off current ratio of about 10^8 and higher current density of $70 \mu\text{A}/\mu\text{m}$ was obtained. The degradation of device DC performance is often found in literatures while converting the substrate from rigid to flexible substrate with a similar fabrication process [35, 50]. Which can be attributed to substrate roughness and poor thermal conductivity that could partly degrade the charge transport properties in these atomically thin materials [51].

Fig. R18. a, The transfer characteristics of MoS₂ FET on PI. The switching ratio (I_{on}/I_{off}) is about 10^8 . b, Output curves, I_{ds} - V_{ds} , for the same device. The device has a current density of $40 \mu\text{A}/\mu\text{m}$ at a V_{ds} of 4 V.

12. Can the authors perform DC and RF measurements of the flexible bilayer MoS₂ devices under various bending conditions? This is the main advantage of using flexible substrates so it is instructional to know if the MoS₂ material can operate under bending conditions.

As the reviewer suggested, DC and RF performance of flexible MoS₂ transistors under various bending conditions were performed. First, the I_dV_g and RF characteristics measurements under cyclic bending with respect to the number of bending cycles (n=0, 10, 100, 1000) were performed. The cyclic bending radius was fixed at 10 cm. The transfer characteristics and normalized f_T of flexible MoS₂ transistors remain stable after 1000 cycles of bending. Furthermore, the I_dV_g and RF characteristics measurements with four different bending radii (flat, R=10, 5, 2 cm) were performed. The bending cycles was fixed at 1000. The $I_{ds}V_{gs}$ curves and f_T of the bilayer MoS₂ transistors with bending radii of 10, 5, and 2 cm degrades slightly compared with that under flat condition. We have added the DC and RF results under bending conditions in the revised Supplementary Fig. 21.

Fig. R19. (a) and (b) display the $I_{ds}V_{gs}$ and normalized f_T of the flexible bilayer MoS₂ transistors after different bending cycles with the bending radius keep at 10 cm. (c) and (d) display the I_dV_g and normalized f_T after four different bending conditions: flat, R= 10 cm 1000 cycles, R= 5 cm 1000 cycles, R= 2 cm 1000 cycles.

13. The study reports on the measured extrinsic values for f_T and f_{max} . Why was the de-embedding procedure performed in the other 2D RF papers not done in this study? The de-embedded f_T and f_{max} parameters give an idea of the intrinsic

material quality, as well as the parasitic effects of the device geometry and composition. It is highly suggested to the authors that the de-embedded values, at least for the best device data, is determined.

a. With the intrinsic f_T value, you can calculate the material velocity saturation value, $v_{sat} = 2\pi \cdot f_T \cdot \text{int} \cdot L$. This is a useful metric for comparing RF FETs as it takes into account gate length. Calculate this value for the best intrinsic f_T data achieved, and compare with the theoretical vsat value for bilayer MoS₂. Include this data column in the supplemental Table 2 of reported results.

We thank the reviewer for detailed and constructive comments. The de-embedding procedure performed in 2D RF papers use various open structures and results in intrinsic de-embedding and standard de-embedding [52]. Intrinsic de-embedded f_T of 78 GHz and f_{max} of 34 GHz are obtained for the best devices data. Intrinsic de-embedding process not only removes the electrode pads but also metal interconnects (and their associated capacitances) in the device structure. While the intent of this de-embedding approach is to determine the intrinsic performance of the active FET channel, we were hesitating of this de-embedding which sometimes overestimate the real performance by removing parasitic related to components of the device required for integration into a practical circuit and result in overestimation of the cut-off frequencies.

As the reviewer suggested, intrinsic de-embedded f_T of 78 GHz are obtained for the best 90 nm device data as shown in Fig. R20, and saturation velocity of 4.4×10^6 cm/s is demonstrated. The saturation velocity have been added in the revised supplemental Table 2 and this saturation velocity is higher than the v_{sat} of 3.4×10^6 cm/s demonstrated in monolayer MoS₂ reported recently[53]. Here, we would like to emphasize that $f_{T,int}$ only represents the upper limit of the possible frequency for this transistor. In any practical applications, the C_{gs} and C_{gd} of the device always significantly affect the device performance. We have added the de-embedded RF results in the revised manuscript and Supplementary Note 15.

Fig. R20. RF characterizations of CVD bilayer MoS₂ RF transistors after intrinsic de-embedding. a. Short-circuit current gain h_{21} of the 90 nm channel length device after intrinsic de-embedding. b. Linear fitting using Gummel's method, showing extrapolated intrinsic cut-off frequency consistent with the intercept value.

14. In Fig. 5b,e label the RF, LO, IF, and other intermodulated harmonic signals.

We thank the reviewer for the valuable comments. The RF, LO, IF and other intermodulated harmonic signals has been labeled in the revised manuscript of Fig. 5b,e.

15. In Fig 5c the mixer shows linear IF gain with LO power from 3 to 8 dBm.

a. Mention this linearity range in the text.

The linearity range of bilayer MoS₂ mixer on rigid and flexible substrates have been added in the revised manuscript.

b. The mixer on flexible substrates in Fig. 5f shows linearity from the full measured range of 3 to 9 dBm. Comment on why the flexible mixer shows better linearity.

We thank the reviewer for pointing out this important question. The original rigid devices are

measured using a RF signal generator as LO power from a domestic vendor (Ceyear AV1464B) where the cable/connector loss is larger at high power output range which causes the fluctuations of the last two data points at 9 and 10 dBm. During the flexible substrate measurement, we used the Agilent network analyzer as the LO power source for RF signals and the cable/connector loss is smaller, and more stable. We have re-measured two more rigid devices also using Agilent network analyzer as LO power to provide a better comparison, and the results on rigid substrates show similar linearity as the flexible ones from 3 to 9 dBm. What's more, better conversion gain of -30.7 were demonstrated at the LO power of 9 dBm for the newly measured devices. We have replaced Fig.5b and 5c with the updated results in the revised manuscript.

Fig. R21. Conversion gain Vs LO power for three devices on rigid substrates.

c. Also in Fig. 5f, if possible measure higher LO power IF gain data points for the flexible device to see when it becomes non-linear.

Higher LO power were mainly limited by cable/connector loss and the current measurement system at this moment. We will keep this idea as a future direction with further optimization of RF measurement techniques.

16. Can the authors include a plot of IF conversion gain vs. frequency? At f_{RF} of 1.5 GHz the mixer shows conversion loss of 33.6 dB. It would be interesting to see at what f_{RF} the IF conversion gain is unity. Keep the input signal powers and

the f_{IF} of 100 MHz the same. Repeat for the flexible mixer.

We thank the reviewer for the valuable comments. The plot of IF conversion gain vs. frequency for rigid and flexible mixers have been included in the revised supplementary information. As shown in Fig. R22 (a) and (b), with the input signal powers and f_{IF} of 100 MHz the same, conversion gain decreases as frequency increase from 1 to 1.7 GHz. In this work, we demonstrated the function of frequency mixer based on MoS₂ transistors without any impedance matching techniques. To achieve unity or positive conversion gain, there still are still many optimizations needed in future such as the matching techniques of the input and output side to achieve maximum power transfer. Also, thinner gate dielectrics to lower the LO power needed will be helpful [54, 55]. We have added the conversion gain vs. frequency results in the revised Supplementary Note 18.

Fig. R22 (a) and (b) IF conversion gain vs. frequency for the rigid and flexible mixers.

17. Please provide a few more details on the test measurement setup for the mixer. Was the test system fully coaxial with SMA connectors? Were there any frequency limitations in the test setup which were overcome? Were impedance matching techniques used?

We thank the reviewer for pointing this out. The mixer measurements are carried out in Lakeshore probe station at room temperature by using an Agilent N5224A network analyzer and Ceyear AV1464B signal generator as input sources, and an Agilent DSA90804A digital (or Agilent N9030B signal analyzer) for the IF signal detection. Bias-Tee (Keysight 11612B) are used both at

the input and the output to combine DC and RF signals, and provide isolation between them. The LO and RF inputs were combined using external power combiner (Keysight 11636C). Coaxial cable with SMA connectors (Rosenberger LA3-C138-100, Rosenberger LU8-C043-1500, SUCOFLEX 101PEA) were used for the signals transmission and the IF signal detection between output bias-tee and signal analyzer. All the instruments, cables, and connectors are meet the frequency requirements for the mixer measurement. It should be noted that none impedance matching techniques were used in this work. The details about the test measurement setup is shown in the Fig. R23. We have added the detail setup in the method section of revised manuscript.

Fig. R23. Measurement setup for MoS₂ FET-based RF mixers

References

- [1] S. Bhaviripudi, X. Jia, M. S. Dresselhaus, and J. Kong, "Role of kinetic factors in chemical vapor deposition synthesis of uniform large area graphene using copper catalyst," *Nano Lett.*, vol. 10, no. 10, pp. 4128-4133, 2010/10/13, 2010.
- [2] S. Zhou, L. Gan, D. Wang, H. Li, and T. Zhai, "Space-confined vapor deposition synthesis of two dimensional materials," *Nano Res.*, vol. 11, no. 6, pp. 2909-2931, 2018/06/01, 2018.
- [3] H. O. Pierson, *Handbook of chemical vapor deposition: principles, technology and applications*: William Andrew, 1999.

- [4] J. Zhou, J. Lin, X. Huang, Y. Zhou, Y. Chen, J. Xia, H. Wang, Y. Xie, H. Yu, J. Lei, D. Wu, F. Liu, Q. Fu, Q. Zeng, C.-H. Hsu, C. Yang, L. Lu, T. Yu, Z. Shen, H. Lin, B. I. Yakobson, Q. Liu, K. Suenaga, G. Liu, and Z. Liu, "A library of atomically thin metal chalcogenides," *Nature*, vol. 556, no. 7701, pp. 355-359, 2018/04/01, 2018.
- [5] S. Wang, Y. Rong, Y. Fan, M. Pacios, H. Bhaskaran, K. He, and J. H. Warner, "Shape evolution of monolayer MoS₂ crystals grown by chemical vapor deposition," *Chemistry of Materials*, vol. 26, no. 22, pp. 6371-6379, 2014/11/25, 2014.
- [6] D. Dumcenco, D. Ovchinnikov, K. Marinov, P. Lazić, M. Gibertini, N. Marzari, O. L. Sanchez, Y.-C. Kung, D. Krasnozhan, M.-W. Chen, S. Bertolazzi, P. Gillet, A. Fontcuberta i Morral, A. Radenovic, and A. Kis, "Large-area epitaxial monolayer MoS₂," *ACS Nano*, vol. 9, no. 4, pp. 4611-4620, 2015/04/28, 2015.
- [7] J. Chen, W. Tang, B. Tian, B. Liu, X. Zhao, Y. Liu, T. Ren, W. Liu, D. Geng, H. Y. Jeong, H. S. Shin, W. Zhou, and K. P. Loh, "Chemical vapor deposition of high-quality large-sized MoS₂ crystals on silicon dioxide substrates," *Advanced Science*, vol. 3, no. 8, pp. 1500033, 2016/08/01, 2016.
- [8] J. Lee, S. Pak, P. Giraud, Y.-W. Lee, Y. Cho, J. Hong, A. R. Jang, H.-S. Chung, W.-K. Hong, H. Y. Jeong, H. S. Shin, L. G. Occhipinti, S. M. Morris, S. Cha, J. I. Sohn, and J. M. Kim, "Thermodynamically stable synthesis of large-scale and highly crystalline transition metal dichalcogenide monolayers and their unipolar n-n heterojunction devices," *Advanced Materials*, vol. 29, no. 33, pp. 1702206, 2017/09/01, 2017.
- [9] K. H. S. Kirby, D. E. Chris, V. S. Saurabh, and P. Eric, "Intrinsic electrical transport and performance projections of synthetic monolayer MoS₂ devices," *2D Materials*, vol. 4, no. 1, pp. 011009, 2017.
- [10] K. Liu, L. Zhang, T. Cao, C. Jin, D. Qiu, Q. Zhou, A. Zettl, P. Yang, S. G. Louie, and F. Wang, "Evolution of interlayer coupling in twisted molybdenum disulfide bilayers," *Nature Communications*, vol. 5, pp. 4966, 09/18/online, 2014.
- [11] H. Ye, J. Zhou, D. Er, C. C. Price, Z. Yu, Y. Liu, J. Lowengrub, J. Lou, Z. Liu, and V. B. Shenoy, "Toward a mechanistic understanding of vertical growth of van der Waals stacked 2D materials: a multiscale model and experiments," *ACS Nano*, vol. 11, no. 12, pp. 12780-12788, 2017/12/26, 2017.

- [12] S. B. Desai, S. R. Madhvapathy, A. B. Sachid, J. P. Llinas, Q. Wang, G. H. Ahn, G. Pitner, M. J. Kim, J. Bokor, C. Hu, H. S. P. Wong, and A. Javey, "MoS₂ transistors with 1-nanometer gate lengths," *Science*, vol. 354, no. 6308, pp. 99, 2016.
- [13] B. Radisavljevic, A. Radenovic, J. Brivio, V. Giacometti, and A. Kis, "Single-layer MoS₂ transistors," *Nat. Nanotechnol.*, vol. 6, pp. 147, 01/30/online, 2011.
- [14] Y.-H. Lee, X.-Q. Zhang, W. Zhang, M.-T. Chang, C.-T. Lin, K.-D. Chang, Y.-C. Yu, J. T.-W. Wang, C.-S. Chang, L.-J. Li, and T.-W. Lin, "Synthesis of large-area MoS₂ atomic layers with chemical vapor deposition," *Adv. Mater.*, vol. 24, no. 17, pp. 2320-2325, 2012/05/02, 2012.
- [15] W. Chen, J. Zhao, J. Zhang, L. Gu, Z. Yang, X. Li, H. Yu, X. Zhu, R. Yang, D. Shi, X. Lin, J. Guo, X. Bai, and G. Zhang, "Oxygen-assisted chemical vapor deposition growth of large single-crystal and high-quality monolayer MoS₂," *J. Am. Chem. Soc.*, vol. 137, no. 50, pp. 15632-15635, 2015/12/23, 2015.
- [16] K. Kang, S. Xie, L. Huang, Y. Han, P. Y. Huang, K. F. Mak, C.-J. Kim, D. Muller, and J. Park, "High-mobility three-atom-thick semiconducting films with wafer-scale homogeneity," *Nature*, vol. 520, pp. 656, 04/30/online, 2015.
- [17] H. Yu, M. Liao, W. Zhao, G. Liu, X. J. Zhou, Z. Wei, X. Xu, K. Liu, Z. Hu, K. Deng, S. Zhou, J.-A. Shi, L. Gu, C. Shen, T. Zhang, L. Du, L. Xie, J. Zhu, W. Chen, R. Yang, D. Shi, and G. Zhang, "Wafer-scale growth and transfer of highly-oriented monolayer MoS₂ continuous films," *ACS Nano*, vol. 11, no. 12, pp. 12001-12007, 2017/12/26, 2017.
- [18] J. Zheng, X. Yan, Z. Lu, H. Qiu, G. Xu, X. Zhou, P. Wang, X. Pan, K. Liu, and L. Jiao, "High-mobility multilayered MoS₂ flakes with low contact resistance grown by chemical vapor deposition," *Adv. Mater.*, vol. 29, no. 13, pp. 1604540, 2017.
- [19] K. Kaasbjerg, K. S. Thygesen, and K. W. Jacobsen, "Phonon-limited mobility in n-type single-layer MoS₂ from first principles," *Phys. Rev. B*, vol. 85, no. 11, pp. 115317, 03/23/, 2012.
- [20] R. Fivaz, and E. Mooser, "Mobility of charge carriers in semiconducting layer structures," *Phys. Rev.*, vol. 163, no. 3, pp. 743-755, 11/15/, 1967.
- [21] X. Cui, G.-H. Lee, Y. D. Kim, G. Arefe, P. Y. Huang, C.-H. Lee, D. A. Chenet, X. Zhang, L. Wang, F. Ye, F. Pizzocchero, B. S. Jessen, K. Watanabe, T. Taniguchi, D. A. Muller, T. Low, P. Kim, and J. Hone, "Multi-terminal transport measurements of MoS₂ using a van der Waals heterostructure device platform," *Nat. Nanotechnol.*, vol. 10, pp. 534, 04/27/online, 2015.

- [22] H. Schmidt, S. Wang, L. Chu, M. Toh, R. Kumar, W. Zhao, A. H. Castro Neto, J. Martin, S. Adam, B. Özyilmaz, and G. Eda, "Transport properties of monolayer MoS₂ grown by chemical vapor deposition," *Nano Lett.*, vol. 14, no. 4, pp. 1909-1913, 2014/04/09, 2014.
- [23] B. Radisavljevic, and A. Kis, "Mobility engineering and a metal–insulator transition in monolayer MoS₂," *Nat. Mater.*, vol. 12, pp. 815, 06/23/online, 2013.
- [24] Z. Yu, Y. Pan, Y. Shen, Z. Wang, Z.-Y. Ong, T. Xu, R. Xin, L. Pan, B. Wang, L. Sun, J. Wang, G. Zhang, Y. W. Zhang, Y. Shi, and X. Wang, "Towards intrinsic charge transport in monolayer molybdenum disulfide by defect and interface engineering," *Nat. Commun.*, vol. 5, pp. 5290, 10/20/online, 2014.
- [25] J. Chen, X. Zhao, S. J. R. Tan, H. Xu, B. Wu, B. Liu, D. Fu, W. Fu, D. Geng, Y. Liu, W. Liu, W. Tang, L. Li, W. Zhou, T. C. Sum, and K. P. Loh, "Chemical vapor deposition of large-size monolayer MoSe₂ crystals on molten glass," *J. Am. Chem. Soc.*, vol. 139, no. 3, pp. 1073-1076, 2017/01/25, 2017.
- [26] P. Yang, X. Zou, Z. Zhang, M. Hong, J. Shi, S. Chen, J. Shu, L. Zhao, S. Jiang, X. Zhou, Y. Huan, C. Xie, P. Gao, Q. Chen, Q. Zhang, Z. Liu, and Y. Zhang, "Batch production of 6-inch uniform monolayer molybdenum disulfide catalyzed by sodium in glass," *Nat. Commun.*, vol. 9, no. 1, pp. 979, 2018/03/07, 2018.
- [27] M. Einax, W. Dieterich, and P. Maass, "Colloquium: cluster growth on surfaces: densities, size distributions, and morphologies," *Rev. Mod. Phys.*, vol. 85, no. 3, pp. 921-939, 07/08/, 2013.
- [28] K. Liu, L. Zhang, T. Cao, C. Jin, D. Qiu, Q. Zhou, A. Zettl, P. Yang, S. G. Louie, and F. Wang, "Evolution of interlayer coupling in twisted molybdenum disulfide bilayers," *Nat. Commun.*, vol. 5, pp. 4966, 09/18/online, 2014.
- [29] H. Lyu, H. Wu, J. Liu, Q. Lu, J. Zhang, X. Wu, J. Li, T. Ma, J. Niu, and W. Ren, "Double-balanced graphene integrated mixer with outstanding linearity," *Nano Lett.*, vol. 15, no. 10, pp. 6677-6682, Sep. 2015.
- [30] Q. Gao, X. Li, M. Tian, X. Xiong, Z. Zhang, and Y. Wu, "Short-channel graphene mixer with high linearity," *IEEE Electron Dev. Lett.*, vol. 38, no. 8, pp. 1168-1171, 2017.
- [31] R. Zeng, P. Li, Y. Wang, G. Wang, Q. Zhang, Y. Liao, and X. Xie, "An embedded gate graphene field effect transistor with natural Al oxidization dielectrics and its application to frequency doubler," *IEICE Electronics Express*, vol. 14, no. 20, pp. 20170707-20170707, 2017.

- [32] O. Habibpour, Z. S. He, W. Strupinski, N. Rorsman, T. Ciuk, P. Ciepielewski, and H. Zirath, "A W-band MMIC Resistive Mixer Based on Epitaxial Graphene FET," *IEEE Microwave and Wireless Components Letters*, vol. 27, no. 2, pp. 168-170, 2017.
- [33] M. Tian, X. Li, T. Li, Q. Gao, X. Xiong, Q. Hu, M. Wang, X. Wang, and Y. Wu, "High-Performance CVD Bernal-Stacked Bilayer Graphene Transistors for Amplifying and Mixing Signals at High Frequencies," *ACS Applied Materials & Interfaces*, vol. 10, no. 24, pp. 20219-20224, 2018/06/20, 2018.
- [34] A. Sanne, R. Ghosh, A. Rai, M. N. Yogeesh, S. H. Shin, A. Sharma, K. Jarvis, L. Mathew, R. Rao, D. Akinwande, and S. Banerjee, "Radio frequency transistors and circuits based on CVD MoS₂," *Nano Lett.*, vol. 15, no. 8, pp. 5039-5045, 2015/08/12, 2015.
- [35] H.-Y. Chang, M. N. Yogeesh, R. Ghosh, A. Rai, A. Sanne, S. Yang, N. Lu, S. K. Banerjee, and D. Akinwande, "Large-area monolayer MoS₂ for flexible low-power RF nanoelectronics in the GHz regime," *Adv. Mater.*, vol. 28, no. 9, pp. 1818-1823, 2016.
- [36] M. S. Gupta, "Power gain in feedback amplifiers, a classic revisited," *IEEE Transactions on Microwave Theory and Techniques*, vol. 40, no. 5, pp. 864-879, 1992.
- [37] J. Lee, S. Pak, P. Giraud, Y.-W. Lee, Y. Cho, J. Hong, A. R. Jang, H.-S. Chung, W.-K. Hong, Y. Jeong Hu, S. Shin Hyeon, G. Occhipinti Luigi, M. Morris Stephen, S. Cha, I. Sohn Jung, and M. Kim Jong, "Thermodynamically stable synthesis of large-scale and highly crystalline transition metal dichalcogenide monolayers and their unipolar n-n heterojunction devices," *Advanced Materials*, vol. 29, no. 33, pp. 1702206, 2017/09/01, 2017.
- [38] K. D. Holland, A. U. Alam, N. Paydavosi, M. Wong, C. M. Rogers, S. Rizwan, D. Kienle, and M. Vaidyanathan, "Impact of contact resistance on the f_T and f_{max} of graphene versus MoS₂ transistors," *IEEE Trans. Nanotechnol.*, vol. 16, no. 1, pp. 94-106, 2017.
- [39] S. J. Han, S. Oida, K. A. Jenkins, D. Lu, and Y. Zhu, "Multifinger embedded T-shaped gate graphene RF transistors with high f_{max}/f_T ratio," *IEEE Electr. Dev. Lett.*, vol. 34, no. 10, pp. 1340-1342, 2013.
- [40] B. W. H. Baugher, H. O. H. Churchill, Y. Yang, and P. Jarillo-Herrero, "Intrinsic electronic transport properties of high-quality monolayer and bilayer MoS₂," *Nano Lett.*, vol. 13, no. 9, pp. 4212-4216, 2013/09/11, 2013.

- [41] S. Y. Chou, and D. A. Antoniadis, "Relationship between measured and intrinsic transconductances of FET's," *IEEE Trans. Electron Devices*, vol. 34, no. 2, pp. 448-450, 1987.
- [42] S. M. Sze, and K. K. Ng, *Physics of semiconductor devices*: John wiley & sons, 2006.
- [43] C. J. McClellan, E. Yalon, K. K. H. Smithe, S. V. Suryavanshi, and E. Pop, "Effective n-type doping of monolayer MoS₂." pp. 1-2.
- [44] A. Rai, A. Valsaraj, H. C. P. Movva, A. Roy, E. Tutuc, L. F. Register, and S. K. Banerjee, "Interfacial-oxygen-vacancy mediated doping of MoS₂ by high-κ dielectrics." pp. 189-190.
- [45] G. Yongji, L. Bo, Y. Gonglan, Y. Shize, Z. Xiaolong, L. Sidong, J. Zehua, B. Elisabeth, V. Soumya, I. Y. Boris, L. Jun, V. Robert, Z. Wu, and M. A. Pulickel, "Direct growth of MoS₂ single crystals on polyimide substrates," *2D Materials*, vol. 4, no. 2, pp. 021028, 2017.
- [46] Z. Liu, M. Amani, S. Najmaei, Q. Xu, X. Zou, W. Zhou, T. Yu, C. Qiu, A. G. Birdwell, F. J. Crowne, R. Vajtai, B. I. Yakobson, Z. Xia, M. Dubey, P. M. Ajayan, and J. Lou, "Strain and structure heterogeneity in MoS₂ atomic layers grown by chemical vapour deposition," *Nat. Commun.*, vol. 5, pp. 5246, 11/18/online, 2014.
- [47] S. McDonnell, B. Brennan, A. Azcatl, N. Lu, H. Dong, C. Buie, J. Kim, C. L. Hinkle, M. J. Kim, and R. M. Wallace, "HfO₂ on MoS₂ by atomic layer deposition: adsorption mechanisms and thickness scalability," *ACS Nano*, vol. 7, no. 11, pp. 10354-10361, 2013/11/26, 2013.
- [48] H. G. Kim, and H.-B.-R. Lee, "Atomic layer deposition on 2D materials," *Chemistry of Materials*, vol. 29, no. 9, pp. 3809-3826, 2017/05/09, 2017.
- [49] T. Li, M. Tian, S. Li, M. Huang, X. Xiong, Q. Hu, S. Li, X. Li, and Y. Wu, "Black Phosphorus Radio Frequency Electronics at Cryogenic Temperatures," *Advanced Electronic Materials*, vol. 4, no. 8, pp. 1800138, 2018/08/01, 2018.
- [50] R. Cheng, S. Jiang, Y. Chen, Y. Liu, N. Weiss, H.-C. Cheng, H. Wu, Y. Huang, and X. Duan, "Few-layer molybdenum disulfide transistors and circuits for high-speed flexible electronics," *Nat. Commun.*, vol. 5, pp. 5143, 10/08/online, 2014.
- [51] S. Park, S. H. Shin, M. N. Yogeesh, A. L. Lee, S. Rahimi, and D. Akinwande, "Extremely high-frequency flexible graphene thin-film transistors," *IEEE Electr. Dev. Lett.*, vol. 37, no. 4, pp. 512-515, 2016.

- [52] H. Wang, X. Wang, F. Xia, L. Wang, H. Jiang, Q. Xia, M. L. Chin, M. Dubey, and S.-j. Han, "Black phosphorus radio-frequency transistors," *Nano Lett.*, vol. 14, no. 11, pp. 6424-6429, 2014/11/12, 2014.
- [53] K. K. H. Smithe, C. D. English, S. V. Suryavanshi, and E. Pop, "High-field transport and velocity saturation in synthetic monolayer MoS₂," *Nano Letters*, vol. 18, no. 7, pp. 4516-4522, 2018/07/11, 2018.
- [54] M. A. Andersson, Y. Zhang, and J. Stake, "A 185-215-GHz subharmonic resistive graphene FET integrated mixer on silicon," *IEEE Trans. Microw. Theory Techn.*, vol. 65, no. 1, pp. 165-172, 2017.
- [55] H. Madan, M. Hollander, M. LaBella, R. Cavalero, D. Snyder, J. Robinson, and S. Datta, "Record high conversion gain ambipolar graphene mixer at 10GHz using scaled gate oxide." pp. 4.3. 1-4.3. 4.

=====

At last, we sincerely appreciate all the reviewers for valuable and critical comments that significantly improve the quality of our manuscript. Thank you very much.

REVIEWERS' COMMENTS:

Reviewer #1 (Remarks to the Author):

All the queries are addressed satisfactorily. However, some of the structural modifications are required as given in following.

1. In Fig. 4a, the contacts and active material are need to index clearly in inside the scheme.
2. All the scale bar values are need to provide inside the figures.
3. In Fig. 4c, the scale bar and its value are overlapped. Need to correct them.
4. Need to provide the better quality bilayer MoS₂ TEM image in Fig. 2i.

Reviewer #2 (Remarks to the Author):

Authors have done good job on answering my questions. I suggest publication of this manuscript without need any further edits.

Reviewer #3 (Remarks to the Author):

The authors have satisfactorily addressed all the technical questions and comments raised by the reviewers. I strongly recommend the publication of this manuscript in Nature Communications after the authors address the following comments and minor issues:

1. Please ensure you use consistent notations and naming throughout the manuscript. For example, the authors sometimes write 'DC' in some instances and 'Dc' or 'dc' in others. Please write everything as 'DC'. Another example is 'Figure' vs. 'Fig.' in the main text - please use only one of the two. Carefully proofread the manuscript for such notational/naming discrepancies.
2. Carefully proofread the manuscript and supplementary information again for any grammatical and/or sentence structuring errors.
3. The authors should make sure that all the figure labels and figure text is easily legible to the reader. Please check all the font sizes again, especially for Figure 4 and Figure 5.
4. On page 9, Line 182, please write $f_{RF}=1.5$ GHz as $f_{RF} = 1.5$ GHz. Give proper spaces for consistency. Please make such changes throughout the manuscript.
5. On Page 7, Lines 135-136, the authors should also mention that the high ON-current can also be attributed to the high-k doping effect due to the interfacial-oxygen-vacancies in the HfLaO dielectric at the bilayer MoS₂ interface. Also, please write 'on-current', 'off-current' as 'ON-current', 'OFF-current' throughout the manuscript.
6. Since the manuscript highlights issues such as contact resistance, mobility and doping of MoS₂, it will be highly instructive if the authors cite the following review paper that was published recently: <http://www.mdpi.com/2073-4352/8/8/316>. This is a comprehensive review paper and will be very useful for the readers. The authors can cite this at an appropriate place within the manuscript where they talk about the mobility, contact resistance etc. of their bilayer MoS₂.

Point-by-point response to reviewers' comments for manuscript

NCOMMS-18-09196A

The manuscript has been revised along with all the reviewers' comments. Changes made in the manuscript are marked using track changes. It is our belief that the manuscript is substantially improved after making the suggested edits. The responses to the reviewers' comments are list below.

===== Authors' responses to the Reviewers' comments =====

REVIEWERS' COMMENTS:

Reviewer #1 (Remarks to the Author):

All the queries are addressed satisfactorily. However, some of the structural modifications are required as given in following.

Our response: We greatly appreciate the positive comments from the reviewer.

1. In Fig. 4a, the contacts and active material are need to index clearly in inside the scheme.

Our response: We wish to thank the reviewer point this out. The contacts and active material have been labeled in the revised Figure 4a.

2. All the scale bar values are need to provide inside the figures.

Our response: We wish to thank the reviewer point this out. We have formalize the scale bars according to the format requirements of nature communications.

3. In Fig. 4c, the scale bar and its value are overlapped. Need to correct them.

Our response: We wish to thank the reviewer point this out. The format have been corrected in the revised manuscript.

4. Need to provide the better quality bilayer MoS₂ TEM image in Fig. 2i.

Our response: We wish to thank the reviewer point this out. The bilayer MoS₂ TEM image in Figure 2i have been replaced with better quality.

Reviewer #2 (Remarks to the Author):

Authors have done good job on answering my questions. I suggest publication of this manuscript without need any further edits.

Our response: We greatly appreciate the positive comments from the reviewer.

Reviewer #3 (Remarks to the Author):

The authors have satisfactorily addressed all the technical questions and comments raised by the reviewers. I strongly recommend the publication of this manuscript in Nature Communications after the authors address the following comments and minor issues:

Our response: We greatly appreciate the positive comments from the reviewer.

1. Please ensure you use consistent notations and naming throughout the manuscript. For example, the authors sometimes write 'DC' in some instances and 'Dc' or 'dc' in others. Please write everything as 'DC'. Another example is 'Figure' vs. 'Fig.' in the main text - please use only one of the two. Carefully proofread the manuscript for such notational/naming discrepancies.

Our response: We wish to thank the reviewer point this out. The manuscript has been through carefully proofread and the notations are consistent throughout the manuscript in the revised manuscript.

2. Carefully proofread the manuscript and supplementary information again for any grammatical and/or sentence structuring errors.

Our response: We wish to thank the reviewer point this out. The manuscript and supplementary information have been carefully proofread to avoid grammatical or sentence structuring errors.

3. The authors should make sure that all the figure labels and figure text is easily legible to the reader. Please check all the font sizes again, especially for Figure 4 and Figure 5.

Our response: We wish to thank the reviewer point this out. The figure labels and figure text sizes have been increased in the revised manuscript.

4. On page 9, Line 182, please write $f_{RF}=1.5$ GHz as $f_{RF} = 1.5$ GHz. Give proper spaces for consistency. Please make such changes throughout the manuscript.

Our response: We wish to thank the reviewer point this out. The format has been revised throughout the manuscript.

5. On Page 7, Lines 135-136, the authors should also mention that the high ON-current can also be attributed to the high- k doping effect due to the interfacial-oxygen-vacancies in the HfLaO dielectric at the bilayer MoS₂ interface. Also, please write 'on-current', 'off-current' as 'ON-current', 'OFF-current' throughout the manuscript.

Our response: We wish to thank the reviewer point this out. The high- k doping effect have been added in the corresponding location in revised manuscript, and on-current have been written as ON-current throughout the manuscript.

6. Since the manuscript highlights issues such as contact resistance, mobility and doping of MoS₂, it will be highly instructive if the authors cite the following review paper that was published recently: <http://www.mdpi.com/2073-4352/8/8/316>. This is a comprehensive review paper and will be very useful for the readers. The authors can cite this at an appropriate place within the manuscript where they talk about the mobility, contact resistance etc. of their bilayer MoS₂.

Our response: We thank the reviewer for providing the useful reference. We have cited the above reference in the revised manuscript as ref 9.